# INSTANTSWAP: FAST CUSTOMIZED CONCEPT SWAPPING ACROSS SHARP SHAPE DIFFERENCES

**Chenyang Zhu**[1, *]**, Kai Li**[2, *, †]**, Yue Ma**[3, *]**, Longxiang Tang**[1]**, Chengyu Fang**[1]**, Chubin Chen**[1]**,
Qifeng Chen**[3]**, Xiu Li**[1, †]
[1] Tsinghua University     [2] Meta     [3] HKUST
https://instantswap.github.io/

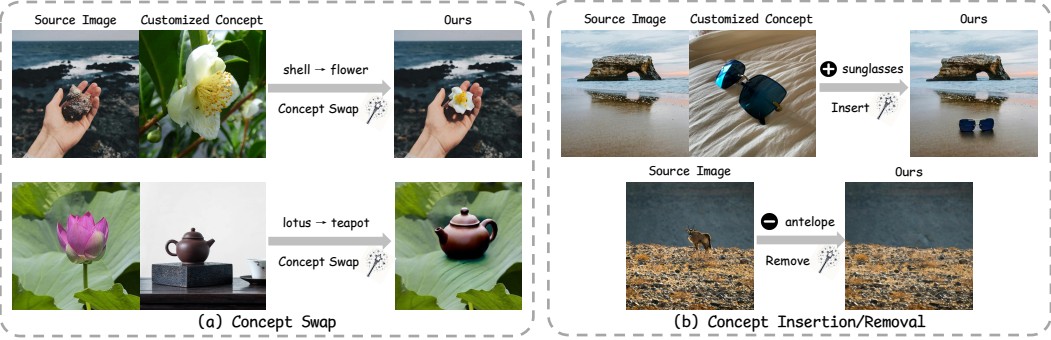

Figure 1: Visual results of INSTANTSWAP. Our approach can seamlessly swap a source concept with a customized concept in an image, even with great shape differences. Moreover, INSTANTSWAP can be used for other tasks, such as concept insertion and removal.

## ABSTRACT

Recent advances in Customized Concept Swapping (CCS) enable a text-to-image model to swap a concept in the source image with a customized target concept. However, the existing methods still face the challenges of ***inconsistency*** and ***inefficiency***. They struggle to maintain consistency in both the foreground and background during concept swapping, especially when the shape difference is large between objects. Additionally, they either require time-consuming training processes or involve redundant calculations during inference. To tackle these issues, we introduce INSTANTSWAP, a new CCS method that aims to handle sharp shape disparity at speed. Specifically, we first extract the bbox of the object in the source image *automatically* based on attention map analysis and leverage the bbox to achieve both foreground and background consistency. For background consistency, we remove the gradient outside the bbox during the swapping process so that the background is free from being modified. For foreground consistency, we employ a cross-attention mechanism to inject semantic information into both source and target concepts inside the box. This helps learn semantic-enhanced representations that encourage the swapping process to focus on the foreground objects. To improve swapping speed, we avoid computing gradients at each timestep but instead calculate them periodically to reduce the number of forward passes, which improves efficiency a lot with a little sacrifice on performance. Finally, we establish a benchmark dataset to facilitate comprehensive evaluation. Extensive evaluations demonstrate the superiority and versatility of INSTANTSWAP.

## 1 INTRODUCTION

We explore the task of Customized Concept Swapping (CCS), a subtask of text-to-image (T2I) generation, which aims to replace a concept in a source image with a highly customized new concept. Combined with diffusion models (Dhariwal & Nichol, 2021; Rombach et al., 2022; Nichol et al., 2021), recent CCS methods demonstrate widespread applicability in areas such as selfie enhancement, photo blog creation, and comic creation.

Early work (Yang et al., 2023) in CCS primarily relies on copy-paste techniques, which are rough and unreliable. By integrating powerful customization techniques (Ruiz et al., 2023; Kumari et al.,

---

* Equal contribution.
† Corresponding Authors.

a kitten playing pool with balls → a kitten playing pool with a sks candle

Figure 2: Our INSTANTSWAP achieves better swapping consistency than the existing methods.

2023) with image editing methods (Hertz et al., 2022; 2023), a series of works (Gu et al., 2024a; Choi et al., 2023; Li et al., 2023b; Gu et al., 2024b) have been proposed. Although achieving remarkable success, these approaches still face the problems of *inconsistency* and *inefficiency* as shown in Fig. 2. (1) *Inconsistency*: Attention-based methods such as PhotoSwap (Gu et al., 2024a) and P2P (Hertz et al., 2022) maintain background consistency well but struggle with shape differences between source and target concepts, resulting in foreground inconsistency. Score distillation based methods such as SDS (Poole et al., 2022), DDS (Hertz et al., 2023), and CDS (Nam et al., 2024) fail to generate foreground concepts precisely and alter the background significantly, causing both foreground and background inconsistency. (2) *Inefficiency*: Attention-based methods require an inefficient training phase (Mokady et al., 2023; Ju et al., 2024) on source image to maintain background consistency. While score distillation based methods are training-free, they still require redundant calculations of forward passes at each timestep, leading to inference inefficiency.

To address the aforementioned issues, we propose INSTANTSWAP, a training-free framework that *efficiently* performs customized concept swapping across shape differences while maintaining ***both foreground and background consistency***. Specifically, we extract the bounding box (bbox) that indicates the position of the source concept from the enhanced cross-attention map of the source image. With this bbox, we perform the background gradient masking (BGM) strategy to prevent modifications outside the bbox, thus ensuring background consistency. Moreover, to improve foreground consistency, we leverage the semantic information to highlight the cross-attention maps of source and target concepts respectively within the bbox. This strategy leads to the semantic-enhanced concept representation (SECR), which facilitates precise foreground swapping. Finally, we introduce the Step-Skipping Gradient Updating (SSGU) strategy, which only performs forward passes at certain timesteps to calculate gradients. For the timesteps without direct gradient computations, we reuse the previously obtained gradients for updates. Through this strategy, we reduce the total number of forward passes and improve the efficiency of our method.

Since the CCS is a recently proposed task, no dedicated evaluation benchmark currently exists. To address this gap, we introduce *ConSwapBench*, the first benchmark dataset specifically designed for CCS. *ConSwapBench* comprises two sub-benchmarks: ConceptBench and SwapBench. Concept-Bench contains images representing target concepts, while SwapBench includes images with one or more concepts to be swapped, serving as source images.

Through extensive qualitative and quantitative comparisons, we demonstrate the effectiveness and superiority of our INSTANTSWAP. We also conduct comprehensive ablation studies to verify the effectiveness of each component of our approach. Additionally, we further extend our INSTANTSWAP to related tasks, proving its efficacy and versatility. Our contributions are summarized as follows:

- We propose INSTANTSWAP, a novel training-free customized concept swapping (CCS) framework, which enables efficient concept swapping across sharp shape differences.
- We design the background gradient masking (BGM) strategy and semantic-enhanced concept representation (SECR) to improve the background and foreground consistency respectively. Moreover, we adopt a step-skipping gradient updating (SSGU) strategy to reduce redundant computation and improve efficiency.
- To provide a comprehensive evaluation for CCS, we introduce *ConSwapBench*, the first benchmark for customized concept swapping. Extensive qualitative and quantitative evaluations demonstrate the effectiveness and superiority of our INSTANTSWAP.

## 2 RELATED WORK

### 2.1 DIFFUSION-BASED IMAGE EDITING

Image Editing is a fundamental and popular topic in computer vision. Previous works based on Generative Adversarial Networks (GAN) (Goodfellow et al., 2020; Li et al., 2020; He et al.; 2023) only

focus on specific object domains (Tang et al., 2023a;b; Fang et al., 2024; He et al., 2024a;c;b; Zhong et al., 2024b;c;a), which limits the application. With the emergence of diffusion model (Rombach et al., 2022), image editing is now able to modify various objects through prompts. These methods are mainly divided into five categories: instruction-based methods, blending-based, attention-based, inversion-based, and score distillation based methods. Instruction-based methods (Brooks et al., 2023; Ma et al., 2024a;b; 2022; 2023; Geng et al., 2024; Huang et al., 2024; Zhang et al., 2024a) typically require an instruction editing dataset to train the diffusion model. Blending-based methods (Couairon et al., 2022; Li et al., 2024; Zhang et al., 2023; Huang et al., 2023a) merge the source and target prompts to guide the editing process, while attention-based methods (Hertz et al., 2022; Cao et al., 2023; Ma et al., 2024c; Tumanyan et al., 2023; Guo & Lin, 2024; Wang et al., 2024a;b) inject the attention feature of the source image. Both methods have lower editing costs but poorer background preservation and prompt alignment. Inversion-based methods (Mokady et al., 2023; Ju et al., 2024; Miyake et al., 2023; Dong et al., 2023; Chen et al., 2024) aim to reverse the fixed trajectory generated by the forward pass to reproduce the source image. These methods can serve as an extra training phase to enhance the background consistency of attention-based methods. Finally, score distillation based methods (Hertz et al., 2023; Nam et al., 2024; Wang et al., 2024a; Feng et al., 2024; Xue et al., 2024; Chang et al., 2024; Kim et al., 2023) draw on the optimization process of SDS (Poole et al., 2022), using score distillation-based loss to optimize the source image for editing. These methods are more flexible than the previous ones but still face challenges with background preservation.

## 2.2 CONCEPT SWAPPING

Concept swapping, a subtask of general image editing, focuses on replacing the source concept in an image with a user-specified target concept. This task is first proposed by PbE (Yang et al., 2023), which employs a CLIP encoder to extract features of the target concept and inject them into the UNet through a cross-attention layer. After that, concurrent works (Gu et al., 2024a; Choi et al., 2023; Li et al., 2023b) extend concept swapping into the customization field (Gal et al., 2022; Zhu et al., 2024). They combine attention-based editing methods (Hertz et al., 2022; Tumanyan et al., 2023), with tuning-based customization methods (Ruiz et al., 2023; Kumari et al., 2023) to achieve customized concept swapping. Building on Photoswap, SwapAnything (Gu et al., 2024b) further obtains masks with external modules to specify the locations of objects in the source image. We improved the existing method in three key aspects. Firstly, we employ bounding boxes instead of masks as spatial indicators of the source concept, allowing greater flexibility for shape variation during concept swapping. Secondly, we use bounding boxes to prevent background changes via gradient masking, thus ensuring background consistency. Third, we augment concept representation with semantic information to maintain foreground consistency. Finally, rather than executing forward passes at every timestep, we execute them only at specific intervals to enhance efficiency.

## 3 METHOD

Given a set of images (typically fewer than 5) $\mathcal{X}_t = \{x_i\}_{i=1}^M$ representing a specific concept $O_t$, along with an image $x_s$ and a prompt $p_s$ describing a source concept $O_s$, the objective of CCS is to "seamlessly" replace $O_s$ in $x_s$ with $O_t$ according to a target prompt $P_t$, resulting in a final target image $x_t$. An ideal customized concept swapping should handle the shape differences between source and target concepts to preserve swapping consistency while maintaining satisfactory efficiency. We introduce INSTANTSWAP to achieve this. INSTANTSWAP is based on Stable Diffusion and extends from the score distillation based image editing methods (Poole et al., 2022; Hertz et al., 2023).

### 3.1 PRELIMINARIES

#### 3.1.1 STABLE DIFFUSION

In this paper, the foundational model utilized for text-to-image generation is Stable Diffusion (Rombach et al., 2022). It takes a text prompt $P$ as input and generates the corresponding image $x$. Stable Diffusion consists of three main components: an autoencoder($\mathcal{E}(\cdot), \mathcal{D}(\cdot)$), a CLIP text encoder $\tau(\cdot)$ and a U-Net $\epsilon_\phi(\cdot)$. Typically, it is trained with the guidance of the following reconstruction loss:

$$\mathcal{L}_{rec} = \mathbb{E}_{z,\epsilon\sim\mathcal{N}(0,1),t,P}\left[\|\epsilon - \epsilon_\phi\left(z_t, t, \tau\left(P\right)\right)\|_2^2\right], \tag{1}$$

where $\epsilon \sim \mathcal{N}(0,1)$ is a randomly sampled noise, t denotes the time step. The calculation of $z_t$ is given by $z_t = \alpha_t z + \sigma_t \epsilon$, where the coefficients $\alpha_t$ and $\sigma_t$ are provided by the noise scheduler.

### 3.1.2 SCORE DISTILLATION BASED IMAGE EDITING

Different from traditional attention-based image editing, score distillation based methods achieve image editing through iterative optimization with a score distillation loss. Given the latent feature $z$ of source image and a denoising U-Net $\epsilon_\phi(\cdot)$, SDS (Poole et al., 2022) can optimize the latent feature $z$ of the image to align with the target prompt $P_t$ by employing the following loss:

$$\mathcal{L}_{SDS} = \|\epsilon_\phi\left(z_t, t, \tau\left(P_t\right)\right) - \epsilon\|_2^2, \tag{2}$$

where $\epsilon$ and $t$ are randomly sampled noise and timestep.

The resulting image SDS is very blurry and only contains foreground objects in the target prompt $P_t$. To address this issue, DDS (Hertz et al., 2023) expresses the gradient of Eq. (2) as

$$\nabla_z \mathcal{L}_{SDS}(z_t, t, \tau(P_t)) = \delta_{tgt} + \delta_{bias}, \tag{3}$$

where $\delta_{tgt}$ indicates the direction aligned with the target prompt and $\delta_{bias}$ refers to undesired part that makes the image blurry. Based on this, DDS further utilizes the fixed latent $\hat{z}_t$ of the source image and the source prompt $P_s$ to approximate the bias component in Eq. (3):

$$\nabla_z \mathcal{L}_{SDS}(\hat{z}_t, t, \tau(P_s)) \approx \hat{\delta}_{bias} \approx \delta_{bias}. \tag{4}$$

Finally, DDS is represented by the difference of Eq. (3) and Eq. (4):

$$\nabla_z \mathcal{L}_{DDS} = \nabla_z \mathcal{L}_{SDS}(z_t, t, \tau(P_t)) - \nabla_z \mathcal{L}_{SDS}(\hat{z}_t, t, \tau(P_s)) \approx \delta_{tgt}. \tag{5}$$

Based on Eq. (5), the loss of DDS is given by

$$\mathcal{L}_{DDS} = \|\epsilon_\phi\left(z_t, t, \tau\left(P_t\right)\right) - \hat{\epsilon}_\phi(\hat{z}_t, t, \tau(P_s))\|_2^2. \tag{6}$$

### 3.2 INSTANTSWAP

Directly extending score distillation based editing methods to the task of CCS encounters the challenge of inconsistency. These methods optimize the background and foreground simultaneously, causing cross-interference and leading to undesirable inconsistency. To address these limitations, we first propose a strategy to *automatically* locate objects to be edited, resulting in the object bounding box (bbox). With this bbox, we propose a background gradient masking technique to remove gradients in the background region and confine swapping to the foreground region. To further enhance foreground swapping consistency, we propose to learn semantic-enhanced concept representations for both source and target concepts based on an attention map feature injection mechanism. An overview of our method is presented in Fig. 3.

### 3.2.1 AUTOMATIC BOUNDING BOX GENERATION

We first automatically obtain the bbox to indicate the position of the concept $O_s$ in the source image. Given the source image $x_s$ and the source prompt $P_s$, we perform a forward pass with the U-Net $\epsilon_\phi(\cdot)$ and obtain the cross-attention map $A^c$ and self-attention map $A^s$ through:

$$A = \text{Softmax}\left(\frac{QK^T}{\sqrt{d'}}\right)V, \tag{7}$$

where $Q$ is the query vector projected from the image features, $d'$ represents the output dimension of key and query features. $K$ is the key vector and $V$ is the value vector. For cross-attention maps $A^c$, $K$ and $V$ are projected from the text embeddings $\tau(P)$. For self-attention maps $A^s$, $K$, and $V$ are projected from the image features. Directly applying a threshold on the $A^c$ can yield a coarse-grained mask, which cannot accurately reflect the location of $O_s$. Inspired by (Nguyen et al., 2024; Zhang et al., 2024b; Tang et al., 2024), we modify the $A^c$ as follows:

$$\hat{A}^c = A^s \cdot (A^c)^\alpha. \tag{8}$$

Based on Eq. (7), all values in $A^c$ range between 0 and 1. Therefore, element-wise exponentiation of $A^c$ by $\alpha$ can weaken the activation of non-target regions. Additionally, as mentioned in (Liu et al., 2024), $A^s$ contains rich structural information. This information can effectively assist $\hat{A}^c$ in better activating the target regions. Finally, we apply the threshold $\beta$ to $\hat{A}^c$ to obtain the mask. Subsequently, we converted the mask into the bbox $B_s$ based on the minimum and maximum coordinates of all foreground points within the mask. This strategy allows us to obtain the bbox $B_s$ without any additional modules. We intentionally set a relatively loose constraint on the mask to obtain a bbox that completely covers the source concept. We discuss the effectiveness of our automatically obtained bboxes in Sec. 4.5.

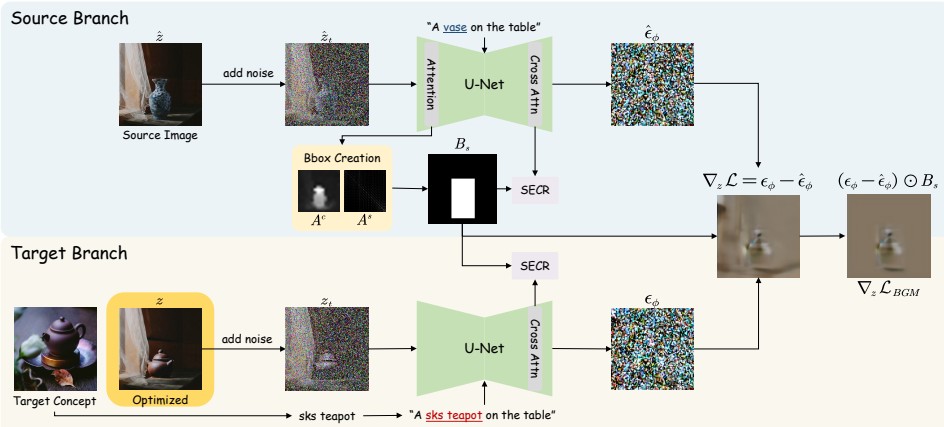

Figure 3: Overall pipeline of INSTANTSWAP. We first obtain the bbox of the source concept automatically. The obtained bbox is input into SECR in both the source and target branches to enhance the foreground swapping consistency. Additionally, the source and target branches generate the prediction of noise for the source and target images based on their respective prompts. The predicted noise, along with the bbox, is used for the BGM to preserve background consistency.

### 3.2.2 BACKGROUND GRADIENT MASKING

With the object bbox, we propose a background gradient masking (BGM) approach to ensure that the concept swap is confined to the foreground region. Given the latent feature $\hat{z}$ of the source image and the latent feature $z$ of the target image, where $z$ is initialized to $\hat{z}$ and is continuously optimized to obtain the final target image $x_t$. Based on Eq. (6), we first obtain the gradient of $z$:

$$\nabla_z \mathcal{L} = (\epsilon_\phi \left( z_t, t, \tau \left( P_t \right) \right) - \hat{\epsilon}_\phi(\hat{z}_t, t, \tau(P_s))) \frac{\partial \epsilon_\phi \left( z_t, t, \tau \left( P_t \right) \right)}{\partial z_t} \frac{\partial z_t}{\partial z}. \tag{9}$$

As stated in (Poole et al., 2022), the mid term is a U-Net Jacobian term and can be omitted, and $\alpha_t = \partial z_t / \partial z$ is a constant which can be represented as $w(t)$:

$$\nabla_z \mathcal{L} = w(t)(\epsilon_\phi \left( z_t, t, \tau \left( P_t \right) \right) - \hat{\epsilon}_\phi(\hat{z}_t, t, \tau(P_s))). \tag{10}$$

This gradient shares the same dimension as $z$, which means it can update $z$ in a pixel-wise manner. However, this will update the foreground and background simultaneously, producing inconsistent background. To remedy this, we apply the bbox $B_s$ on Eq. (10) to mask the gradients related to the background before back propagation and obtain our BGM:

$$\nabla_z \mathcal{L}_{BGM} = w(t)(\epsilon_\phi \left( z_t, t, \tau \left( P_t \right) \right) - \hat{\epsilon}_\phi(\hat{z}_t, t, \tau(P_s))) \odot B_s. \tag{11}$$

This simple masking strategy prevents the background from being updated and thus ensures background consistency.

### 3.2.3 SEMANTIC-ENHANCED CONCEPT REPRESENTATION

The BGM module maintains the background consistency during swapping. However, whether the source concept can be replaced with the target concept cannot be guaranteed. This limitation arises because the optimization of Eq. (11) is still carried out at the entire feature map level of both the source latent $\hat{z}_t$ and target latent $z_t$ without distinguishing between the foreground and the background. To address this, we propose to obtain semantic-enhanced concept representations for both source and target concepts and emphasize their locations within the foreground region during concept swapping.

Let $F_s$ be the source image feature and $p_s$ represent the prompt of source concept (e.g. "rose"), the semantic embedding $c_s$ can be acquired through $c_s = \tau(p_s)$. We first resize the previously obtained object bbox to fit the dimensions of the source image feature $F_s$, resulting in the feature bbox $B_f$. We then crop $F_s$ with $B_f$ to get a regional image feature $f_s$. With $f_s$, we calculate the query vector through $Q_s = W^q \cdot f_s$. After that, we can obtain the key and value vectors through:

$$K_s = W^k \cdot c_s, V_s = W^v \cdot c_s. \tag{12}$$

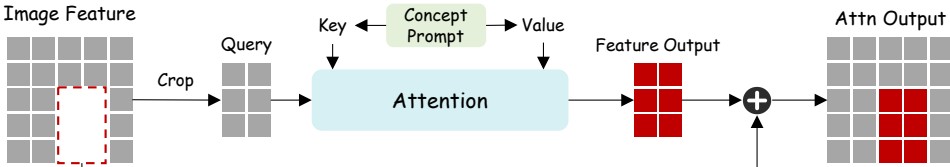

Figure 4: Overview of SECR.

Then the final partial attention output is calculated as follows:

$$\hat{f}_s = \text{Softmax}\left(\frac{Q_s K_s^T}{\sqrt{d'}}\right) V_s, \tag{13}$$

where $d'$ represents the output dimension of key and query features. In this way, we inject the semantic information of the source concept into the cross-attention map, resulting in regional concept representation $\hat{f}_s$. We then map $\hat{f}_s$ back to the original feature map $F_s$ to get a semantic-enhanced representation $\hat{F}_s$ for the entire source image. Fig. 4 illustrates the process. In the target branch, we first convert the target concept into semantic space with DreamBooth (see more details in Appendix N), using a specific rare token (e.g., "sks") to represent the concept. With the target prompt $p_t$ (e.g., "sks teapot") and the feature bbox $B_f$, we similarly apply this process for the target image feature $F_t$ and obtain the semantic-enhanced representation $\hat{F}_t$ for the target image.

Through proactive injection of semantic guidance, we provide the source and target branches with semantic-enhanced concept representation within the foreground region. Consequently, SECR transforms the target branch into a target concept adder and the source branch into a source concept remover. Their collaboration results in precise and seamless concept swapping, thus enhancing the foreground consistency. Moreover, SECR can also facilitate concept insertion and removal, which is further discussed in Sec. 4.6.

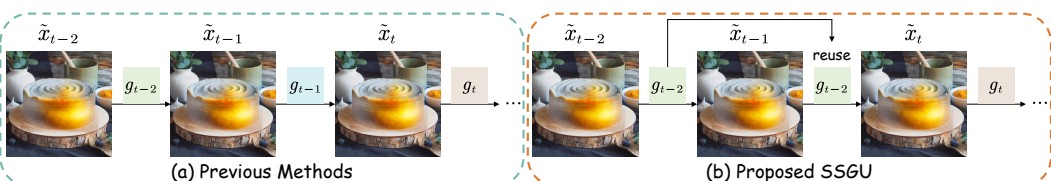

Figure 5: Comparison between our SSGU and previous methods.

### 3.2.4 STEP-SKIPPING GRADIENT UPDATING

After addressing the problem of inconsistency, we turn our attention to the challenge of inefficiency. As illustrated in Fig. 5, previous methods calculate the gradient at each timestep. However, the success of DDIM (Song et al., 2020) in accelerating DDPM (Ho et al., 2020) motivates us to consider: *Can we skip the calculation of gradients at certain timesteps?* During concept swapping, we observe that the effect of gradient updates on the target image is similar across adjacent timesteps (see detailed results in Appendix I). Based on this observation, we propose the Step-Skipping Gradient Updating (SSGU) strategy. The key insight of SSGU is that *skipping some gradient calculations does not significantly sacrifice the swapping consistency while considerably improving efficiency*. As a result, our SSGU calculates gradients at interval timesteps and reuses the previously calculated gradients during the intervening timesteps.

We define our entire pipeline as $\mathcal{F}$, given the source image $x_s$, the timestep $t$, and the intermediate target image $\tilde{x}_t$ at timestep $t$. We can obtain the gradient $g_t$ and the output intermediate target image $\tilde{x}_{t+1}$ at timestep $t$ as follows:

$$g_t = \mathcal{F}(x_s, \tilde{x}_t, t), \tag{14}$$
$$\tilde{x}_{t+1} = \tilde{x}_t - \eta g_t, \tag{15}$$

where $\eta$ is the learning rate. Our SSGU periodically retains some anchor gradients and skips the forward passes between two anchor gradients. The step-skipping period is controlled by the SSGU factor $\lambda$. The set of anchor gradients can be defined as:

$$\mathbb{G} = \{g_{\lambda k}\}, k = 0, 1 \cdots, \lfloor T/\lambda \rfloor, \tag{16}$$

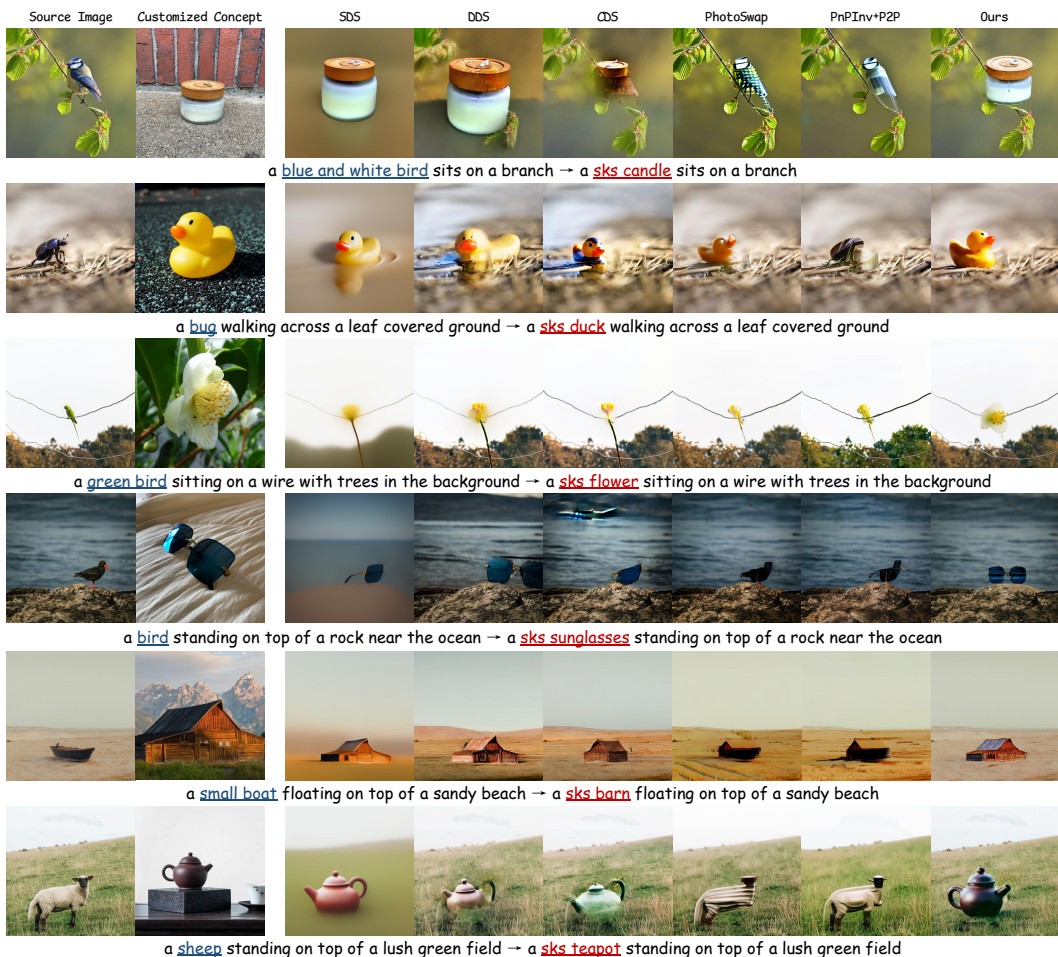

Figure 6: Qualitative comparisons between our INSTANTSWAP and other methods. More qualitative results as well as the used bboxes can be found in Appendix Q.

where $T$ is the ended timestep. For any intervening timestep $t$, we use its nearest former anchor gradient to update the intermediate target image $\tilde{x}_t$. Taking $\lambda = 2$ as an example, we assume that $t$ is an even number and $g_{t-2}, g_t \in \mathbb{G}$. SSGU updates $\tilde{x}_{t-2}$ and $\tilde{x}_{t-1}$ with the anchor gradient $g_{t-2}$:

$$g_{t-2} = \mathcal{F}(x_s, \tilde{x}_{t-2}, t-2), \tag{17}$$

$$\tilde{x}_{t-1} = \tilde{x}_{t-2} - \eta g_{t-2}, \tag{18}$$

$$\tilde{x}_t = \tilde{x}_{t-1} - \eta g_{t-2}. \tag{19}$$

For the next timestep $t$, another anchor gradient $g_t$ is used to update $x_t$. As a result, our SSGU reduces the number of forward passes during the entire concept swapping process to $1/\lambda$ of the original count. Since the forward pass accounts for approximately 95% of the total inference time (see detailed analysis in Appendix E), our SSGU can improve the overall inference speed of our method by approximately $\lambda$ times, with minimal effect on swapping consistency (see Sec. 4.5). Furthermore, our SSGU can be transferred to other score distillation based methods to improve their efficiency in the same way, which is further discussed in Sec. 4.6.

## 4 EXPERIMENTS

### 4.1 IMPLEMENTATION DETAILS

We conduct the experiments with Stable Diffusion (Rombach et al., 2022) v2.1-base on a single RTX3090. We use the customized checkpoint from DreamBooth (Ruiz et al., 2023) to introduce concepts. We set the SSGU factor $\lambda$ to 5, $\alpha$ to 2, $\beta$ to 0.5 and the guidance scale to 7.5. The bbox is obtained through the first three steps. Subsequently, we use SGD (Robbins & Monro, 1951) with a learning rate of 0.1 to optimize for 550 steps of iterations.

Table 1: Quantitative comparisons. Our method outperforms all the compared methods in all the selected metrics. **Red** stands for the best result, **Blue** stands for the second best result.

| Method | FG | BG | | | | Overall | |
|---|---|---|---|---|---|---|---|
| | CLIP-I $\uparrow$ | PSNR $\uparrow$ | LPIPS $_{\times 10^3} \downarrow$ | MSE $_{\times 10^4} \downarrow$ | SSIM $_{\times 10^2} \uparrow$ | CLIP-T $\uparrow$ | Time (s) $\downarrow$ |
| SDS | **73.70** | 20.79 | 339.51 | 107.53 | 72.59 | 23.53 | 40.37 |
| DDS | 71.05 | 24.07 | **89.80** | 53.08 | **83.44** | 23.99 | 66.89 |
| CDS | 71.69 | 23.36 | 90.35 | 63.41 | 83.21 | 24.17 | 140.26 |
| PhotoSwap | 70.15 | 24.24 | 120.62 | 56.64 | 80.56 | 22.38 | 140.34 |
| PnPInv+P2P | 70.74 | **24.63** | 108.22 | **47.49** | 82.07 | **24.25** | **37.02** |
| Ours | **75.00** | **27.39** | **47.68** | **27.87** | **86.58** | **25.74** | **19.83** |

## 4.2 CONSWAPBENCH

Despite the significant application potential of customized concept swapping, there is currently no dedicated evaluation benchmark. To meet the needs of comprehensive evaluation, we introduce *ConSwapBench*, the first benchmark dataset specifically designed for customized concept swapping. *ConSwapBench* consists of two sub-benchmarks: ConceptBench and SwapBench. ConceptBench comprises 62 images covering 10 different target concepts used for customization, while SwapBench includes 160 real images containing one or more objects to be swapped, serving as source images. For each image in SwapBench, we use Grounding SAM (Ren et al., 2024) to acquire the bbox of the foreground concepts as the ground truth for evaluation purposes. We apply each customized concept from ConceptBench to perform concept swaps on each image in SwapBench, ultimately generating a total of 1,600 images for evaluation. More details can be found in Appendix C.

## 4.3 QUALITATIVE COMPARISON

Since customized concept swapping is a relatively novel task, there are limited methods available for direct comparison. Consequently, we include SOTA image editing methods and adapt them for customized concept swapping. We include the following methods: (1) *Score distillation based* methods: SDS (Poole et al., 2022), DDS (Hertz et al., 2023), and CDS (Nam et al., 2024); (2) *Attention-based* methods: PhotoSwap (Gu et al., 2024a), PnPInv (Ju et al., 2024), and P2P (Hertz et al., 2022). We excluded SwapAnything (Gu et al., 2024b) as it is not publicly available. The qualitative results are illustrated in Fig. 6. We find that score distillation based methods can accommodate shape variations during concept swapping. However, they exhibit poor foreground fidelity (3rd and 6th rows) and lead to unnecessary modifications on the background (1st and 2nd rows). Attention-based methods are unable to manage shape variations (4th row) and also struggle with maintaining background consistency (5th row). In contrast, our method demonstrates superior performance in addressing shape variations and maintaining swapping consistency.

## 4.4 QUANTITATIVE COMPARISON

We also conduct a thorough quantitative comparison on *ConSwapBench*. For each generated image, we first use the ground truth bbox in SwapBench to obtain their foreground and background respectively. We use seven different metrics to evaluate the methods from three aspects: (1) Foreground consistency: We calculate the CLIP Image Score (Radford et al., 2021) between the foreground of generated images and the images of customized concepts. (2) Background consistency: We use the four metrics, PSNR, LPIPS (Zhang et al., 2018), MSE, SSIM (Wang et al., 2004) to evaluate the background consistency. (3) Overall consistency and efficiency: We calculate the CLIP Text Score (Hessel et al., 2021) between generated images and target prompts to evaluate the overall prompt consistency. We also report the inference time of each method to evaluate their efficiency. As shown in Tab. 1, our method outperforms other methods on all seven metrics.

## 4.5 ABLATION STUDY

**BGM.** To verify the effectiveness of BGM in background preservation, we conduct an ablation study by removing BGM. As illustrated in the second column of Fig. 7, while our method can still achieve concept swapping without

Table 2: Quantitative ablation results of BGM.

| Method | PSNR $\uparrow$ | LPIPS $\downarrow$ | MSE $\downarrow$ | SSIM $\uparrow$ | CLIP-T $\uparrow$ |
|---|---|---|---|---|---|
| w/o BGM | 18.03 | 249.24 | 184.19 | 72.25 | 23.08 |
| Ours | **27.39** | **47.68** | **27.87** | **86.58** | **25.74** |

BGM, it causes serious modifications on the background. In contrast, our full method not only maintains high foreground fidelity but also effectively preserves the background consistency. We

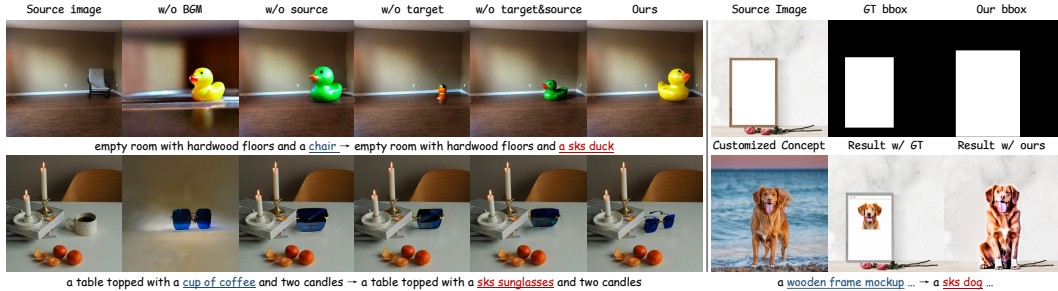

Figure 7: Qualitative results of the ablation study on: *Left*: BGM and SECR. *Right*: different bboxes.

further conduct a quantitative analysis of the background consistency and prompt consistency, as shown in Tab. 2. Our full method outperforms in all metrics.

**Automatic bounding box detection mechanism.** We further verify the effectiveness of our bboxes by using ground truth (GT) bboxes from SwapBench to replace the automatically obtained bboxes. As shown in Fig. 7, although GT bboxes accurately indicate the location of the source concept, they prevent our method from fully swapping the source concept. Compared to GT bboxes, our bboxes are relatively larger and can fully cover the source concept, thereby facilitating complete concept swapping. We also provide quantitative comparisons of different bboxes. As shown in Tab. 3, Gen stands for the generation bboxes, while Eva stands for the evaluation bboxes. The two types of bboxes do not significantly affect background preservation in our method, whereas our bboxes perform better than GT bboxes on the foreground metric. More detailed comparisons can be found in Appendix F.

Table 3: Quantitative ablation results of different bboxes.

| Gen | Eva | CLIP-I ↑ | PSNR ↑ | LPIPS ↓ | MSE ↓ | SSIM ↑ | CLIP-T ↑ |
|-----|-----|----------|--------|---------|-------|--------|----------|
| Ours | GT | 75.00 | 27.39 | 47.68 | 27.87 | 86.58 | **25.74** |
| GT | GT | 75.79 | 31.46 | 32.61 | 13.51 | 87.62 | 25.53 |
| Ours | Ours | **77.72** | **31.64** | **31.17** | **13.28** | **88.21** | **25.74** |

**SECR.** To verify the effectiveness of SECR, we conduct ablation studies including removing SECR from (1) source branch (w/o source), (2) target branch (w/o target), (3) both (w/o source & target). The visualization results are illustrated in columns 3 to 5 of Fig. 7. Although all methods preserve the background well, they show reduced foreground fidelity. Additionally, we perform a quantitative analysis of their foreground consistency and prompt consistency. The results presented in Tab. 4 indicate that our full method exhibits superior performance.

Table 4: Quantitative ablation results of SECR.

| Method | CLIP-I ↑ | CLIP-T ↑ |
|--------|----------|----------|
| w/o source | 73.70 | 25.47 |
| w/o target | 73.40 | 25.52 |
| w/o source&target | 72.42 | 25.21 |
| Ours | **75.00** | **25.74** |

**SSGU.** To verify the effectiveness of our proposed SSGU, we first visualize the images generated under different SSGU factors. As shown in Fig. 9, $\lambda = 1$ indicates that SSGU is not used. When $\lambda \leq 9$, the SSGU can preserve foreground and background consistency well while improving the efficiency of our method. As $\lambda$ increases, the images exhibit more artifacts due to excessive neglect of gradients. Therefore, identifying an optimal SSGU factor $\lambda$ is crucial. We further conduct a detailed quantitative analysis of different $\lambda$ values on foreground consistency and efficiency (see complete results in Appendix D), as illustrated in Fig. 8, where the $x$-axis represents different $\lambda$ values and the $y$-axis represents the respective metric outcomes. When SSGU is not

Figure 8: Quantitative results on the ablation study of the SSGU factor $\lambda$.

used, our method achieves the best swapping consistency but the lowest efficiency. As $\lambda$ increases, our SSGU sacrifices certain swapping consistency but significantly improves efficiency. To balance consistency and efficiency, we ultimately select $\lambda = 5$ for our final model.

### 4.6 APPLICATIONS OF INSTANTSWAP

**Multi-concept swapping.** Our INSTANTSWAP can be easily extended to facilitate multi-concept swaps by sequentially performing multiple single-concept swaps. As shown in the left of Fig. 10, our method can swap each concept within the image with both foreground and background consistency.

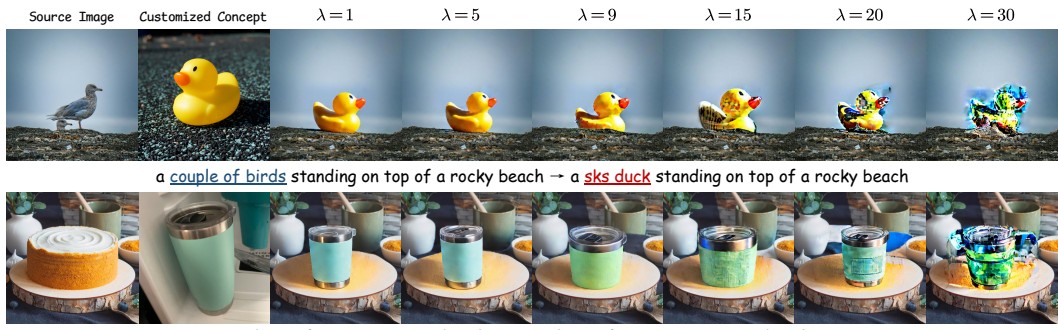

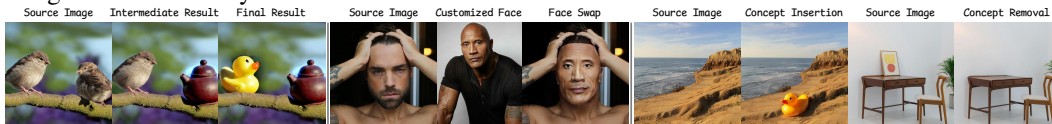

Figure 9: Qualitative results of different SSGU factor $\lambda$. Excessively high $\lambda$ can lead to a decline in foreground consistency.

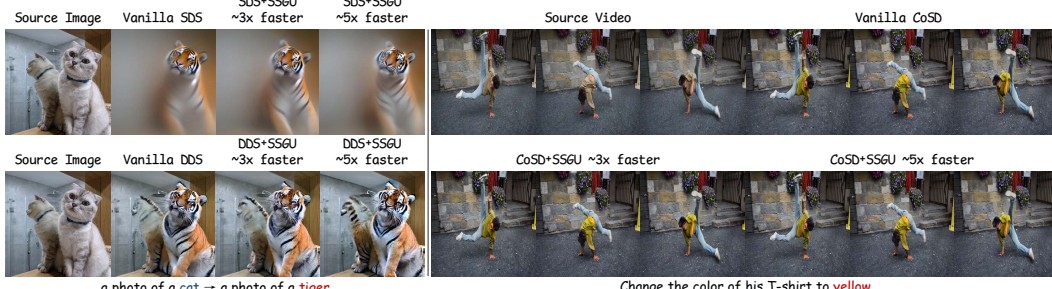

Figure 10: INSTANTSWAP can be extended to other tasks such as: *Left*: Multi-Concept Swapping. *Middle*: Human Face Swapping. *Right*: Concept Insertion and Removal.

**Human face swapping.** INSTANTSWAP demonstrates exceptional capabilities in human face swapping. As shown in the middle of Fig. 10, with customized face models from CivitAI (Civitai, 2024), users can seamlessly replace the face in a source image with a customized target face.

**Concept insertion and removal.** In addition to concept swapping, our method also supports concept insertion and removal. For concept insertion, we employ the same procedure as concept swapping. For concept removal, we adjust the target prompt and the target semantic input $p_t$ of SECR to a null prompt. The results in the right of Fig. 10 further demonstrate the versatility of our method.

**Accelerating other methods.** SSGU can be transferred to other score distillation based methods to enhance their efficiency. We select three representative methods: SDS (Poole et al., 2022), DDS (Hertz et al., 2023) for image editing, and CoSD Kim et al. (2023) for video editing. As shown in Fig. 11, combining these methods with SSGU can significantly improve their efficiency while almost not altering the generation quality. We further conduct a quantitative analysis to assess the transferability of the proposed SSGU, as presented in Tab. 5.

Table 5: Quantitative results of SSGU extension.

| Method | w/o SSGU | w/ SSGU $\lambda = 3$ | w/ SSGU $\lambda = 5$ |
|---|---|---|---|
| SDS | 40.37s | 14.12s | 8.62s |
| DDS | 66.89s | 22.65s | 13.90s |
| CoSD | 344.76s | 128.97s | 79.15s |

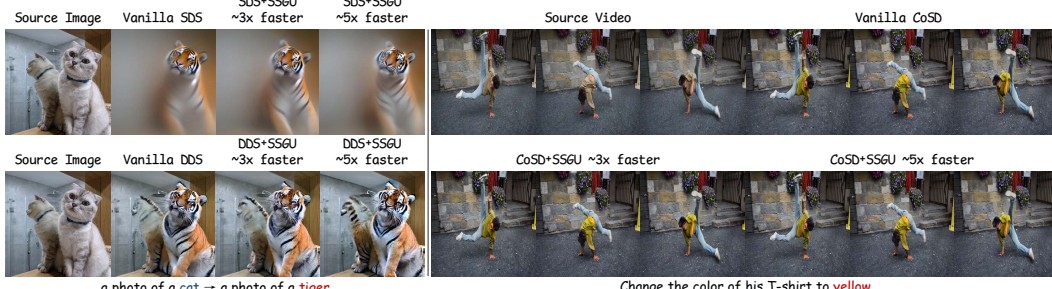

Figure 11: Qualitative results of extending our SSGU to other score distillation based methods.

## 5 CONCLUSION

This paper introduces INSTANTSWAP, a novel framework for precise and efficient customized concept swap. Our BGM and SECR collaborate to maintain both background and foreground consistency. Furthermore, we propose the SSGU to eliminate redundant computation and improve efficiency. Finally, we introduce *ConSwapBench*, a comprehensive benchmark dataset for customized concept swapping. The impressive performance of INSTANTSWAP demonstrates its effectiveness.

**Acknowledgment.** This work was supported by the STI 2030-Major Projects under Grant 2021ZD0201404.

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

## A    REPRODUCIBILITY STATEMENT

To ensure the reproducibility and comprehensiveness of our method, the Appendix includes the following sections: Appendix B offers a detailed explanation of our method, along with the source code and parameter settings for each comparative method. Appendix C reviews relevant benchmarks from previous works and provides detailed information and visual examples of the two sub-benchmarks in *ConSwapBench*. Appendix D provides complete quantitative results of the ablation study on SSGU. Appendix E analyzes the time of each component in our method during the inference process. Appendix F compares our automatically obtained bounding boxes with the ground truth bounding boxes. Appendix G adapts our method to other Stable Diffusion models. Appendix H explores more challenging concepts for concept swapping. Appendix I visualizes the swapping process of our method. Appendix J includes the ablation study of automatic bbox generation. Appendix K discusses the impact of the number of target images.Appendix L shows the time cost of the customization process.Appendix M provides comprehensive results of using Textual Inversion (Gal et al., 2022) for customization. Appendix N explains the detailed usage of the customization methods.Appendix O analyses how our SECR can be applied to DiT-based architecture. Appendix P discusses the impact of our method on the naturalness of generated images. Appendix Q presents the bounding boxes used in the main paper and includes a gallery of additional qualitative results. Appendix R discusses the limitations of our method and future works. In Appendix S, we discuss the potential social impact brought by our method. The code is available in `https://github.com/chenyangzhu1/InstantSwap`.

## B    DETAILED IMPLEMENTATIONS

In this section, we provide experimental details of our method as well as all the comparison methods.

**SDS (Poole et al., 2022).** We utilized a third-party implementation[*] for SDS, adhering to its recommended configurations: an iteration number of 200, a guidance scale of 7.5, and a learning rate of 0.1 with SGD (Robbins & Monro, 1951).

**DDS (Hertz et al., 2023).** We used the official implementation[*] for DDS. We followed its recommended settings, setting the iteration to 200, the guidance scale to 7.5, and using SGD (Robbins & Monro, 1951) with a learning rate of 0.1.

**CDS (Nam et al., 2024).** We utilized the official implementation[†] for CDS, adhering to its recommended settings: an iteration number of 400, a guidance scale of 7.5, and $\omega_{con}$ of 3.0. SGD (Robbins & Monro, 1951) is employed with a learning rate of 0.1.

**PhotoSwap (Gu et al., 2024a).** We utilized the official implementation[‡] of PhotoSwap, adhering to its recommended settings. Specifically, we employed the DDIM (Song et al., 2020) sampling method with 50 denoising steps and a classifier-free guidance scale of 7.5. The default step for cross-attention map replacement is set at 20, while the default steps for self-attention map and feature replacements are 25 and 10, respectively.

**PnPInversion (Ju et al., 2024).** We utilized the official implementation[§] of PnPInversion, adhering to its recommended settings: a step number of 50, a reverse guidance scale of 1, and a forward guidance scale of 0.

**Ours.** We set the SSGU factor $\lambda$ to 5, $\alpha$ to 2 and $\beta$ to 0.5. We use the first three steps to obtain the bbox and then use SGD (Robbins & Monro, 1951) with a learning rate of 0.1 to optimize for 550 steps of iteration. Besides, we follow Huang et al. (2023b) to adopt a non-increasing timestep strategy. To ensure a fair comparison, we set the iteration steps for SDS, DDS, CDS, and our method to the same value when evaluating inference time.

---

[*]https://github.com/google/prompt-to-prompt/blob/main/DDS_zeroshot.ipynb
[†]https://github.com/HyelinNAM/ContrastiveDenoisingScore
[‡]https://github.com/eric-ai-lab/photoswap
[§]https://github.com/cure-lab/PnPInversion

## C   MORE DETAILS OF CONSWAPBENCH

Customized concept swapping is a relatively new task, lacking a mature and comprehensive benchmark for evaluation. Previous approaches have introduced separate benchmarks for image customization and image editing tasks. For image customization, DreamBooth (Ruiz et al., 2023) introduces DreamBench, which includes 30 subjects such as sunglasses, backpacks, dogs, and cats. Custom Diffusion Kumari et al. (2023) proposes CustomConcept101, comprising 101 concepts, including toys, pets, landscapes, and faces. For image editing, PnP (Tumanyan et al., 2023) first introduces a benchmark with 55 image-prompt pairs. PnPInversion (Ju et al., 2024) proposes PIE-Bench, consisting of 700 images, each with five annotations: source prompt, target prompt, editing type, editing content, and corresponding editing mask. Although PIE-Bench covers various editing types, fewer than 80 images are designed for concept swapping, which is insufficient for comprehensive evaluation. To systematically assess our proposed method, we construct a benchmark named *ConSwapBench*, consisting of ConceptBench and SwapBench.

### C.1   CONCEPTBENCH

ConceptBench consists of 62 images representing 10 different concepts from the DreamBench (Ruiz et al., 2023) and CustomConcept101 (Kumari et al., 2023). These concepts include barn, candle, cat, cup, dog, duck, flower, sunglasses, teapot, and vase. We use SAM (Kirillov et al., 2023) to segment the subjects in ConceptBench, ensuring that each target concept is accurately translated into semantic space as customized concepts. We visualize our ConceptBench in Fig. 12.

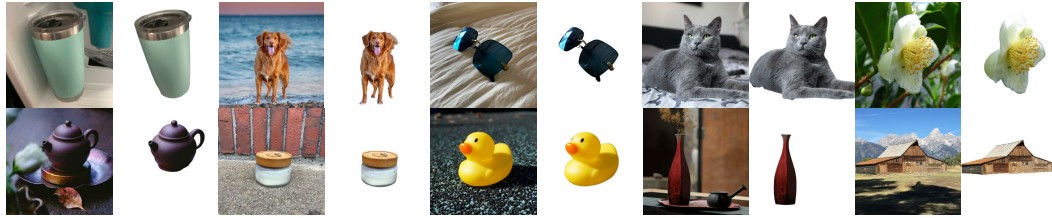

Figure 12: Visualization of ConceptBench.

### C.2   SWAPBENCH

SwapBench comprises 160 images obtained from Unsplash (Unsplash, 2024) and other related benchmarks including PIE-Bench (Ju et al., 2024) and Visual Genome (Krishna et al., 2017). Each image contains one or more objects, serving as the source concept for concept swap. For images with captions, the caption is used as the source prompt. For images without captions, we employ BLIP2 (Li et al., 2023a) to generate a caption. We also use Grounding SAM (Ren et al., 2024) to acquire the bbox of the foreground concepts as the ground truth for evaluation purposes. During the evaluation phase, each concept from ConceptBench is applied to each image in SwapBench for concept swapping, generating 1,600 images for evaluation. We visualize our SwapBench in Fig. 13.

## D   COMPLETE QUANTITATIVE RESULTS OF THE ABLATION STUDY ON SSGU

We provide detailed quantitative results of different $\lambda$ values across all the metrics. As shown in Fig. 14, overall, as $\lambda$ increases, the consistency of our method decreases while its efficiency increases.

## E   TIME ANALYSIS OF DIFFERENT INFERENCE COMPONENTS

The inference process of our method can be divided into three main components: forward pass, backward pass, and other processes. We illustrate the time allocation for these components in our method with the SSGU factor set to 1. As shown in Fig. 15, the forward pass accounts for approximately 95% of the total inference time. Based on this, our SSGU significantly improves the efficiency of our method by eliminating redundant forward passes.

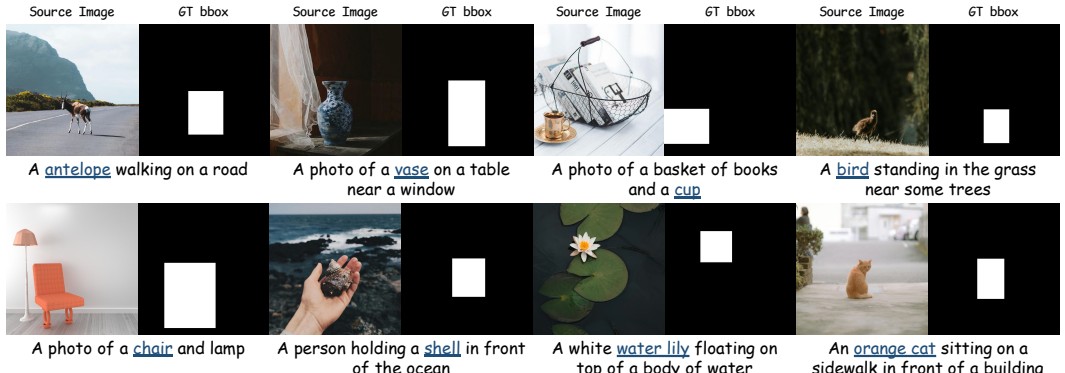

Figure 13: Visualization of SwapBench.

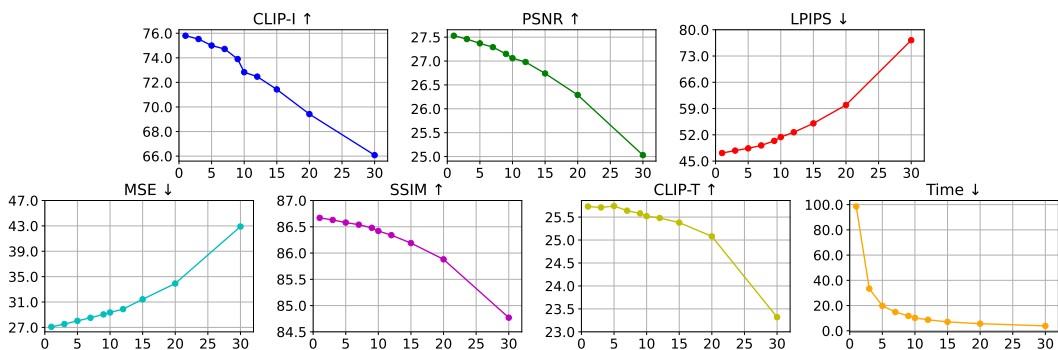

Figure 14: Complete quantitative results on the ablation study of the SSGU factor $\lambda$.

# F    COMPARISON OF DIFFERENT BOUNDING BOXES

In Fig. 16, we show our automatically obtained bounding boxes and the ground truth bounding boxes obtained through Grounding SAM (Ren et al., 2024). Compared to the ground truth bounding boxes, our bounding boxes are larger and can fully cover the source concept. This allows our method to completely swap the source concept with the target concept.

# G    ADAPT INSTANTSWAP TO OTHER STABLE DIFFUSION MODELS

We combine our method with Stable Diffusion v1.5 and conduct comprehensive experiments. We present the quantitative results in Tab. 6 (the inference time metric is obtained on a single A100).

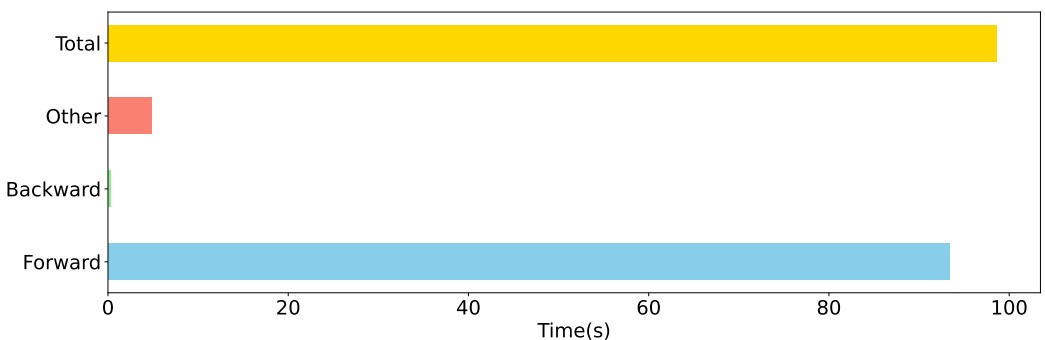

Figure 15: Time allocation of different components during inference.

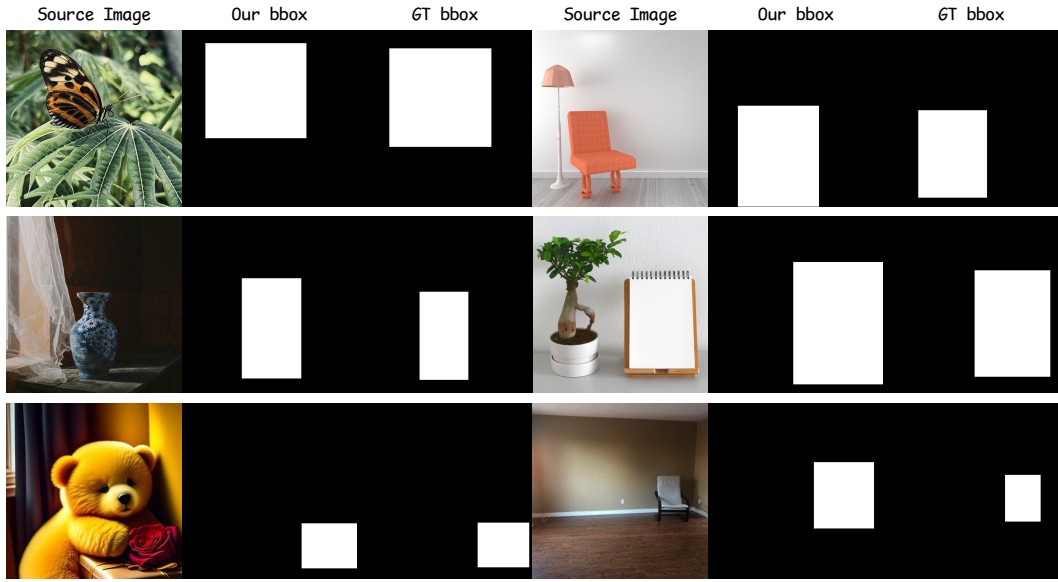

Figure 16: Comparison of the ground truth bounding boxes and our automatically obtained bounding boxes.

We also provide qualitative results in Fig. 17 The results show that our method can integrate well with Stable Diffusion v1.5.

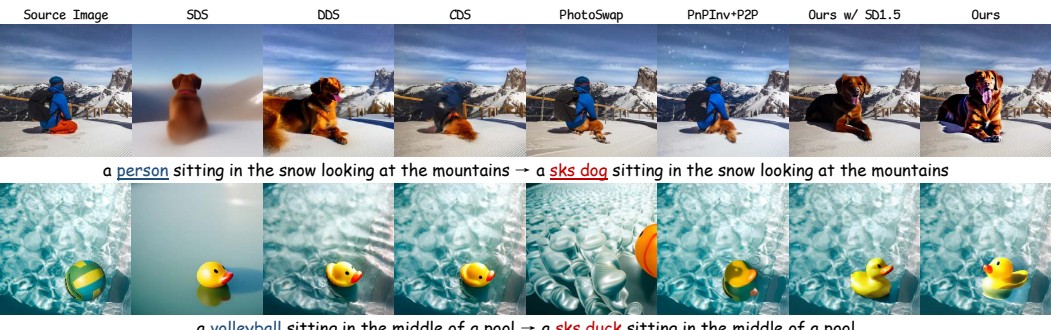

Figure 17: Qualitative results of our method with other Stable Diffusion models.

Table 6: Quantitative results of our method with other Stable Diffusion models.

| Methods | CLIP-I ↑ | PSNR ↑ | LPIPS ↓ | MSE ↓ | SSIM ↑ | CLIP-T ↑ | Inference Time ↓ |
|---|---|---|---|---|---|---|---|
| Ours w/ SD 1.5 | 74.27 | 27.98 | 47.04 | 25.24 | 86.01 | 26.10 | 15.09s |
| Ours2 | 72.70 | 15.25 | 327.23 | 350.01 | 64.80 | 21.23 | 13.38s |

## H  MORE CHALLENGING CONCEPTS

We include more uncommon challenging concepts from DreamBench++ (Peng et al., 2024). We provide qualitative results in Fig. 18. The results show that these challenging concepts indeed reduce the foreground consistency of our method. Nevertheless, our approach still faithfully completes the concept swapping and surpasses all compared methods on all metrics.

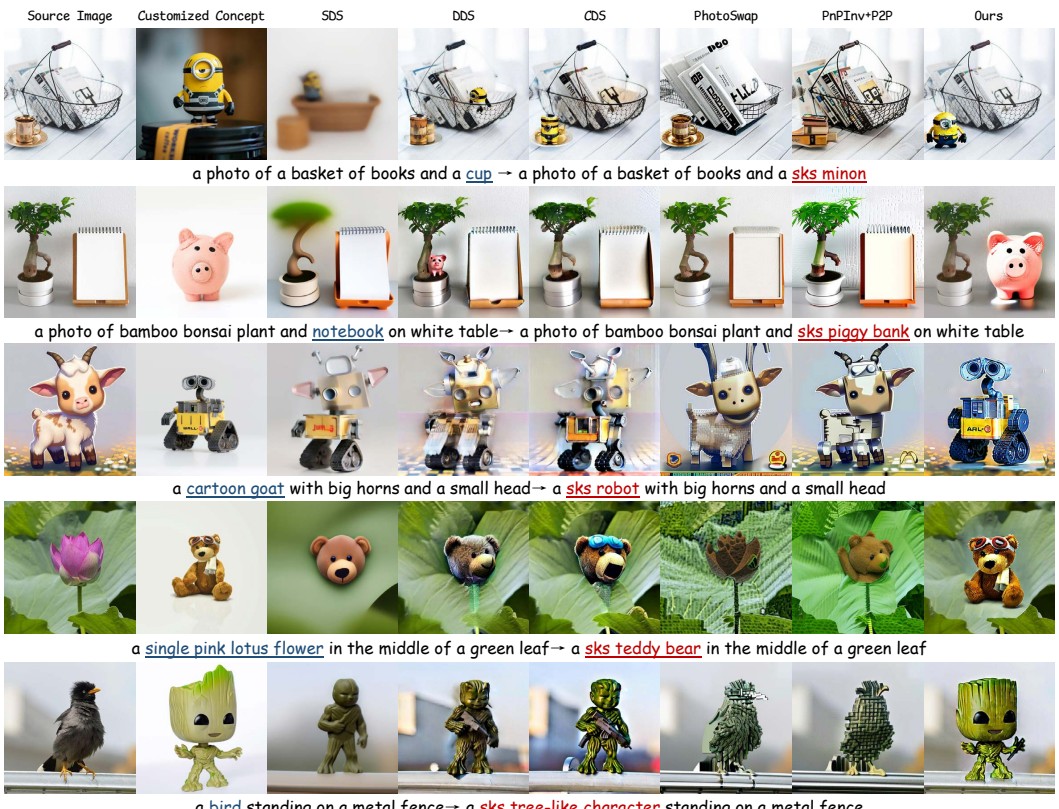

Figure 18: Qualitative results of more challenging concepts.

## I   VISUALIZATION OF THE SWAPPING PROCESS

We visualize the swapping process of our INSTANTSWAP to intuitively demonstrate how it manages shape differences between the source and target concepts. As depicted in Fig. 19, our method can flexibly swap the source concept for the target concept through an optimization approach. Additionally, we visualize the images of each swapping step along with their corresponding gradients. As shown in Fig. 20, we select three different stages of swapping timesteps: 0-4, 100-104, and 200-204 with the SSGU factor set to 1. It can be observed that the gradients at adjacent timesteps update the target image towards a similar direction which does not result in significant changes within each update. This observation supports the use of our proposed SSGU.

## J   ABLATION STUDY ON AUTOMATIC BOUNDING BOX GENERATION

To comprehensively analyze how the combination of self-attention and cross-attention for automatic bbox generation affects the performance, we first set the element-wise exponentiation of $A^s$ and $A^c$ in Eq. (8) to $\alpha_s$ and $\alpha_c$ respectively:

$$\hat{A}^c = (A^s)^{\alpha_s} \cdot (A^c)^{\alpha_c}, \tag{20}$$

where $A^s \in [0,1]^{HW \times HW}$, $A^c \in [0,1]^{HW \times 1}$, $H$ and $W$ are height and weight of the image latent, $A^c$ is the cross-attention map corresponding to the source concept. We apply the threshold to normalized $\hat{A}^c \in \mathbb{R}^{H \times W}$ to obtain the bbox $B_s$. We exclude the trivial case where both $\alpha_s$ and $\alpha_c$ are zero and explore different combinations of self-attention and cross-attention. The qualitative results is shown in Fig. 21. (1) $\alpha_s = 0$, $\alpha_c = 1, 2$: In this case, $A^s$ is an all-ones matrix, so each element in $\hat{A}^c$ has the same value, which is equal to the sum of all elements in $A^c$. After normalization and applying a threshold, the entire bounding box is activated. (2) $\alpha_s = 1, \alpha_c = 0$: In this case, $A^c$ is a vector of all ones, so each item in $\hat{A}^c$ is the sum of the elements in the corresponding

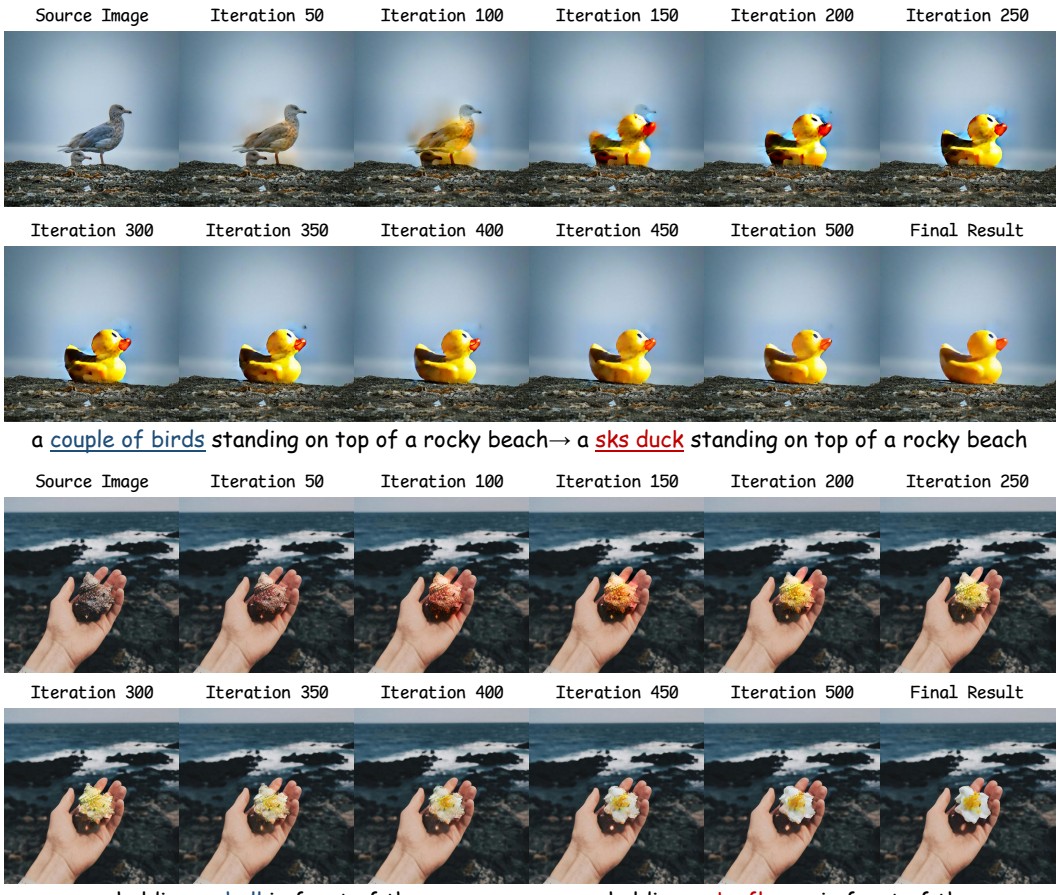

a couple of birds standing on top of a rocky beach→ a <u style="color:red">sks duck standing on top of a rocky beach

a person holding a shell in front of the ocean→ a person holding a <u style="color:red">sks flower in front of the ocean

Figure 19: Visualization of the whole swapping process of INSTANTSWAP.

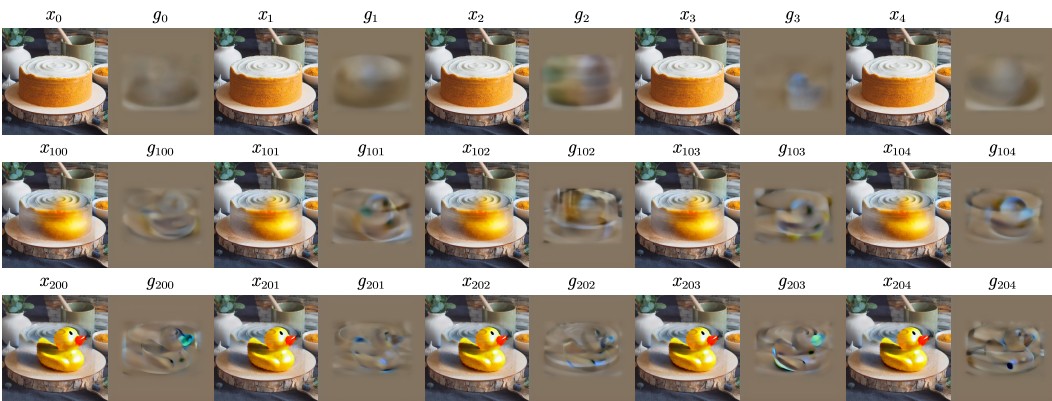

Figure 20: Visualization of the target image and the corresponding gradient at adjacent timesteps.

row of $A^s$. Since $A^s$ is the output of softmax, the sum of each row's elements in $A^s$ is 1. Therefore, $\hat{A}^c$ is also a matrix of all ones, which ultimately results in the entire bounding box being activated. (3) $\alpha_s = 1$, $\alpha_c = 1$: In this case, the target region in the image cannot be highlighted, resulting in a larger bbox, which causes the background of the source image to also be unnecessarily altered. (4) $\alpha_s = 2$, $\alpha_c = 0$: In this situation, using self-attention map alone cannot effectively highlight the foreground region, resulting in a very imprecise bounding box. (5) $\alpha_s = 2$, $\alpha_c = 1, 2$: In this case, the self-attention map takes a leading role in the bbox generation process, producing a smaller

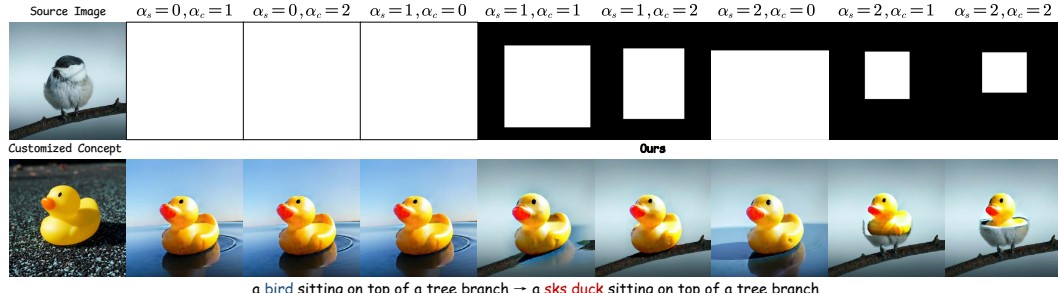

Figure 21: Qualitative results of the ablation study on automatic bounding box generation. We add black borders to the bboxes for better visibility.

bbox, thereby reducing foreground consistency. (6) $\alpha_s = 1$, $\alpha_c = 2$ (Ours): Our setting achieves a proper balance between foreground and background consistency, fully covering the source concept while minimizing background modification. The quantitative in Tab. 7 results in the table further corroborate our analysis.

Table 7: Quantitative ablation results of automatic bbox generation.

| $\alpha_s, \alpha_c$ | CLIP-I ↑ | PSNR ↑ | LPIPS ↓ | MSE ↓ | SSIM ↑ | CLIP-T ↑ |
|---|---|---|---|---|---|---|
| 0, 1 | 72.70 | 15.25 | 327.23 | 350.01 | 64.80 | 21.23 |
| 0, 2 | 72.70 | 15.25 | 327.23 | 350.01 | 64.80 | 21.23 |
| 1, 0 | 72.70 | 15.25 | 327.23 | 350.01 | 64.80 | 21.23 |
| 1, 1 | 74.18 | 23.83 | 93.12 | 82.44 | 82.95 | 25.19 |
| 1, 2 (Ours) | **75.00** | 27.39 | 47.68 | 27.87 | 86.58 | **25.74** |
| 2, 0 | 73.66 | 26.49 | 65.56 | 56.81 | 84.81 | 24.97 |
| 2, 1 | 73.67 | 30.08 | 37.34 | 19.15 | 87.30 | 25.12 |
| 2, 2 | 72.66 | **30.86** | **34.38** | **15.38** | **87.54** | 24.78 |

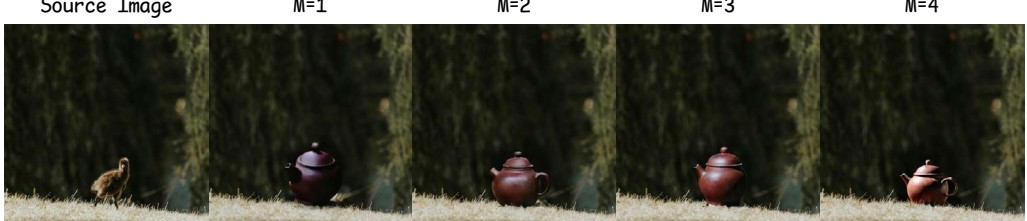

a bird standing in the grass near some trees → a sks teapot standing in the grass near some trees

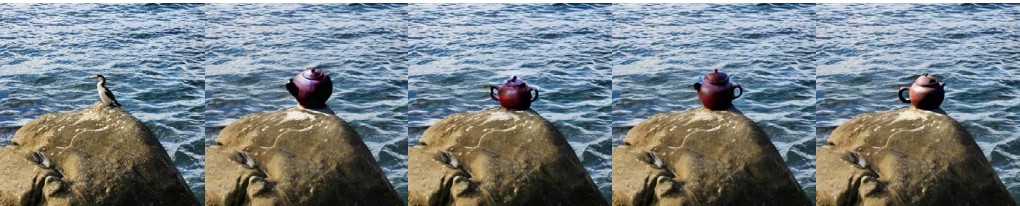

a bird sitting on top of a rock in the water→ a sks teapot sitting on top of a rock in the water

Figure 22: Qualitative results of the ablation study on the number of customization target images.

## K  ANALYSIS OF THE NUMBER OF CUSTOMIZATION TARGET IMAGES

Before concept swapping, our method utilizes DreamBooth (Ruiz et al., 2023) and a set of images (typically fewer than 5) $\mathcal{X}_t = \{x_i\}_{i=1}^M$, for customizing certain concepts, where $M$ is the number of target images used in the customization process. In this section, we discuss how this number $M$ affects the final performance of our method. When customizing the teapot concept our method

can effectively perform concept swapping with just one target image for customization as shown in Fig. 22. As the number of target images increases, the customization becomes more detailed, resulting in more natural outcomes, such as better alignment of shadows and highlights with the lighting angle. Additionally, we conduct quantitative experiments on foreground and prompt consistency metrics. The results in Tab. 8 indicate that the performance of our method improves as $M$ increases.

Table 8: Quantitative ablation results of the number of customization target images.

|  | $M = 1$ | $M = 2$ | $M = 3$ | $M = 4$ |
|---|---|---|---|---|
| CLIP-I ↑ | 74.78 | 75.40 | 75.57 | **75.65** |
| CLIP-T ↑ | 28.94 | 28.93 | 29.13 | **29.44** |

## L    DISCUSSION ON THE TIME COST OF THE CUSTOMIZATION PROCESS

Before concept swapping, our method utilizes DreamBooth (Ruiz et al., 2023) for customizing certain concepts. As shown in Tab. 9, DreamBooth only requires about 8.5 minutes for each concept. Meanwhile, our method is not limited to using DreamBooth for customization. It can be combined with any other more efficient customization methods such as Custom Diffusion (Kumari et al., 2023) which only requires about 3.5 minutes for each concept to improve customization efficiency. We visualize the results of our method combined with Custom Diffusion in Fig. 23.

Table 9: Training time of different customization methods.

|  | DreamBooth (Ruiz et al., 2023) | Custom Diffusion (Kumari et al., 2023) |
|---|---|---|
| Training Time | $\sim 8.5$min | $\sim 3.5$min |

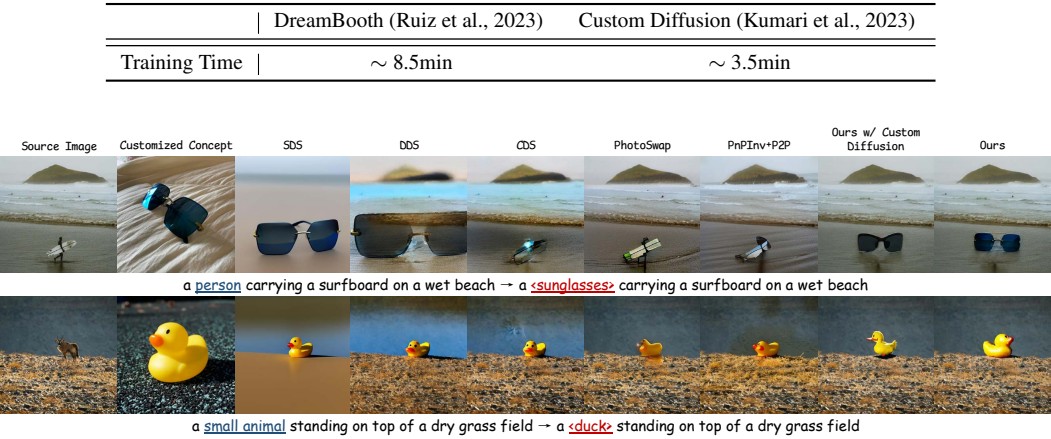

Figure 23: Visualization of our method with Custom Diffusion.

## M    UTILIZING TEXTUAL INVERSION FOR CUSTOMIZATION

Our method is compatible with any customization techniques for concept swapping. To demonstrate its generalization capability, we replace DreamBooth (Ruiz et al., 2023) with Textual Inversion (Gal et al., 2022) and conduct comprehensive experiments. As shown in Fig. 24, our method integrates effectively with Textual Inversion to perform customized concept swapping across shape differences, highlighting its generalization ability. We also provide detailed quantitative results. As shown in the Tab. 10, ours with Textual Inversion achieves similar performance in background consistency and inference time to our original method. Due to the limited customization capabilities of Textual Inversion, its foreground consistency is not as high as with our original method.

## N    DETAILED USAGE OF THE CUSTOMIZATION METHODS

In this section, we detail the use of customization methods in our method. Taking DreamBooth (Ruiz et al., 2023) as an example, given a set of images (typically fewer than five) $\mathcal{X}_t = \{x_i\}_{i=1}^M$ represents

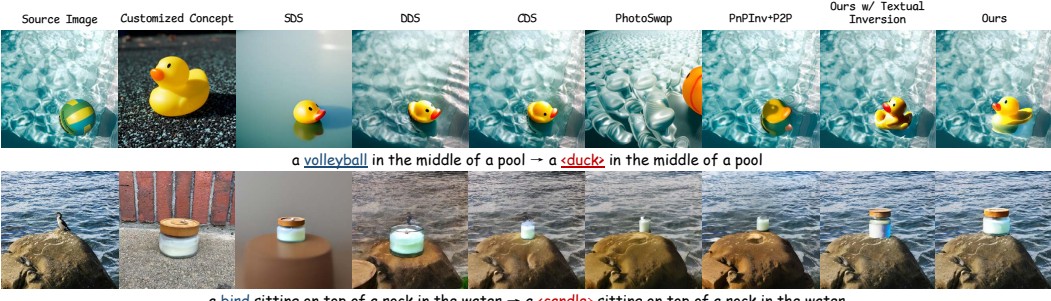

Figure 24: Qualitative results of our method with Textual Inversion.

Table 10: Quantitative results of our method with Textual Inversion.

| Method | FG | BG | | | | Overall | |
|---|---|---|---|---|---|---|---|
| | CLIP-I ↑ | PSNR ↑ | LPIPS$_{\times 10^3}$ ↓ | MSE $_{\times 10^4}$ ↓ | SSIM $_{\times 10^2}$ ↑ | CLIP-T ↑ | Time (s) ↓ |
| Ours w/ Textual Inversion | 72.20 | **28.16** | **45.91** | **24.20** | **86.76** | 24.57 | 19.92 |
| Ours | **75.00** | 27.39 | 47.68 | 27.87 | 86.58 | **25.74** | **19.83** |

a specific concept $O_t$. DreamBooth utilizes these images, combined with a text prompt containing a rare token and the name of $O_t$ (e.g., "`a sks teapot`"), to fine-tune a text-to-image diffusion model. During the finetuning process, DreamBooth is trained using the reconstruction loss of Eq. 1 and a prior preservation loss, which leverages the model's semantic prior to encourage diverse generation results. After customization with DreamBooth, we obtain a checkpoint of the concept $O_t$ and a rare token that semantically represents it. During concept swapping, this checkpoint is loaded, and the rare token is used in the target branch to activate the concept, enabling customized concept swapping. Furthermore, our method can be integrated with other customization methods besides DreamBooth (e.g., Custom Diffusion (Kumari et al., 2023) and Textual Inversion (Gal et al., 2022)). Related discussions can be found in Appendices L and M.

## O APPLY SECR TO DiT-BASED ARCHITECTURES

In our method, SECR leverages semantic information to enhance the cross-attention maps of concepts. This technique can be applied to any generative model with a cross-attention layer to enhance semantic information in regions of interest. However, current DiT-based models, such as SD3 (Esser et al., 2024), typically do not include a cross-attention layer, as the original paper (Peebles & Xie,

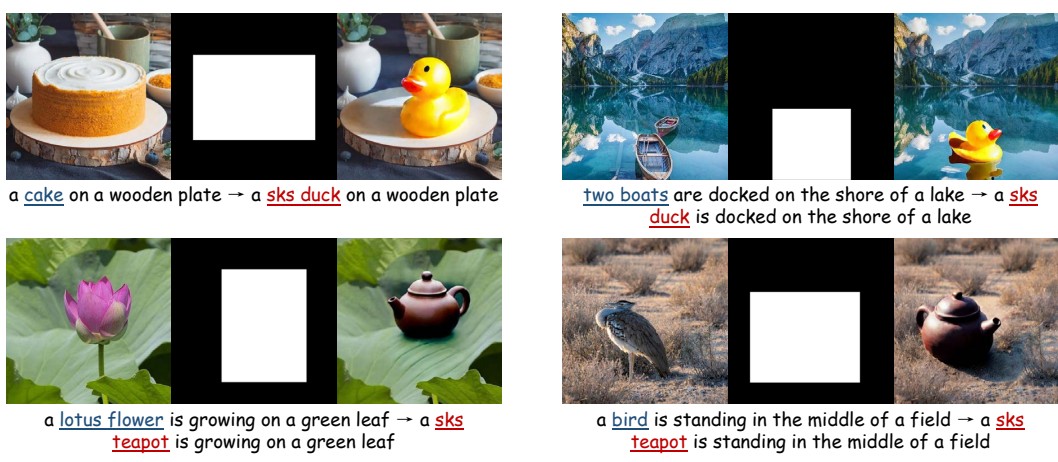

Figure 25: Qualitative results of image naturalness.

Table 11: Quantitative results of image naturalness.

| | SDS | DDS | CDS | PhotoSwap | PnPInv+P2P | Ours |
|---|---|---|---|---|---|---|
| Human Preference Score ↑ | -1.41 | -0.21 | -0.01 | -0.86 | -0.14 | **0.51** |

2023) of DiT shows that using an adaptive layer norm (adaLN) block yields better results than a cross-attention layer. Consequently, most subsequent DiT-based models, including SD3, prioritized using adaLN over employing cross-attention.

Thus, we explore the architecture of SD3 to investigate how our SECR could be applied to SD3. SD3 consists of multiple blocks named MM-DiT Blocks, where each block contains only a self-attention layer. We focus on how images and text interact within these blocks. Within the MM-DiT Block, the image features and text features are processed separately by two separate branches to obtain image latent $z_x$ and text latent $z_t$. Before inputting these latent into the attention layer, the corresponding $Q_x$, $K_x$, $V_x$ of $z_x$ and $Q_t$, $K_t$, $V_t$ of $z_t$ are concatenated together and undergo self-attention as a whole. After the attention layer, the concatenated output is split back into the corresponding $z'_x$ and $z'_t$, returning to their independent branches. Specifically, the input of the attention layer in MM-DiT block can be represented as:

$$Q = \begin{bmatrix} Q_t \\ Q_x \end{bmatrix}, \; K = \begin{bmatrix} K_t \\ K_x \end{bmatrix}, \; V = \begin{bmatrix} V_t \\ V_x \end{bmatrix}. \tag{21}$$

The attention output can be calculated through:

$$\text{Softmax}\left(\frac{QK^T}{\sqrt{d'}}\right) V, \tag{22}$$

where

$$QK^T = \begin{bmatrix} Q_t \\ Q_x \end{bmatrix} \cdot \begin{bmatrix} K_t^T & K_x^T \end{bmatrix} = \begin{bmatrix} Q_t K_t^T & Q_t K_x^T \\ Q_x K_t^T & Q_x K_x^T \end{bmatrix}. \tag{23}$$

Combing Eq. (21), Eq. (22) and Eq. (23), we can obtain the attention output:

$$
\begin{aligned}
\text{Softmax}\left(\frac{QK^T}{\sqrt{d'}}\right) V &= \begin{bmatrix} S\left(Q_t K_t^T\right) & S\left(Q_t K_x^T\right) \\ S\left(Q_x K_t^T\right) & S\left(Q_x K_x^T\right) \end{bmatrix} \cdot \begin{bmatrix} V_t \\ V_x \end{bmatrix} \\
&= \begin{bmatrix} S\left(Q_t K_t^T\right) V_t + S\left(Q_t K_x^T\right) V_x \\ S\left(Q_x K_t^T\right) V_t + S\left(Q_x K_x^T\right) V_x \end{bmatrix},
\end{aligned}
\tag{24}
$$

where $S(\cdot)$ stands for $\text{Softmax}(Q_i K_i^T / \sqrt{d'})$, $i = t, x$. Corresponding to the concatenation method of $Q$, $K$ and $V$, the upper part in Eq. (24) is the text latent output, and the lower part is the image latent output. For image latent output $z'_x$, we have:

$$z'_t = \underbrace{\text{Softmax}\left(\frac{Q_x K_t^T}{\sqrt{d'}}\right) V_t}_{cross\ attention} + \underbrace{\text{Softmax}\left(\frac{Q_x K_x^T}{\sqrt{d'}}\right) V_x}_{self\ attention}. \tag{25}$$

Eq. (25) demonstrates that the attention layer in the MM-DiT block is similar to a cross-atten layer plus a self-attention layer. The first term in Eq. (25) is the cross-attention operation between the image feature and text feature. The section term is the self-attention operation of the image feature. Consequently, our SECR can be applied to the cross-attention part of Eq. (25) to enhance semantic information in regions of interest. However, since SD3 is trained based on rectified flow (Liu et al., 2022; Albergo & Vanden-Eijnden, 2022; Lipman et al., 2022), existing score distillation methods (Poole et al., 2022; Hertz et al., 2023) cannot be directly integrated with it. In future work, we will continue to explore this potential direction.

## P  IMPACT OF SECR ON THE NATURALNESS OF IMAGES

In this section, we discuss the impact of our method on the naturalness of generated images. We apply the SECR strategy to foreground concepts within the cross-attention layers to obtain the

semantic-enhanced representation of the source/target object. This enhanced representation of the foreground region interacts effectively with the features of other surrounding objects and the background in the self-attention layers, mitigating inconsistencies and unnaturalness in the target image. We present comprehensive qualitative and quantitative results to support our analysis. As shown in Fig. 25, although we apply the SECR strategy to the foreground, our method successfully enables the correct interaction of concepts with the whole scene. Specifically: (1) Our method seamlessly inpaints objects in the background that the foreground concept occludes (upper left examples). (2) It generates reflections of the target concept in water (upper right examples). (3) It accurately places the target concept on background objects (lower left examples). (4) It also generates natural shadows of the target concept that are consistent with the environmental lighting angles (upper left and lower right examples). Additionally, considering that image naturalness is a subjective human assessment, we use the human preference metric (Xu et al., 2024) to quantify the naturalness of images. A higher score indicates a better alignment with human preferences. As shown in Tab. 11, our method is the only one that receives a positive human evaluation score, demonstrating a significant advantage over other methods.

## Q   ADDITIONAL QUALITATIVE RESULTS

Due to page limitations, we included only a limited number of visualizations in the main paper. To clearly and intuitively demonstrate the effectiveness of our method, as shown in Fig. 26, we first provide the bounding boxes used in the qualitative comparisons of the main paper. Subsequently, we provide more qualitative comparisons in Figs. 28 and 29. Finally, we present a gallery (see Figs. 30 to 35) of additional qualitative results to comprehensively showcase the performance of our proposed INSTANTSWAP.

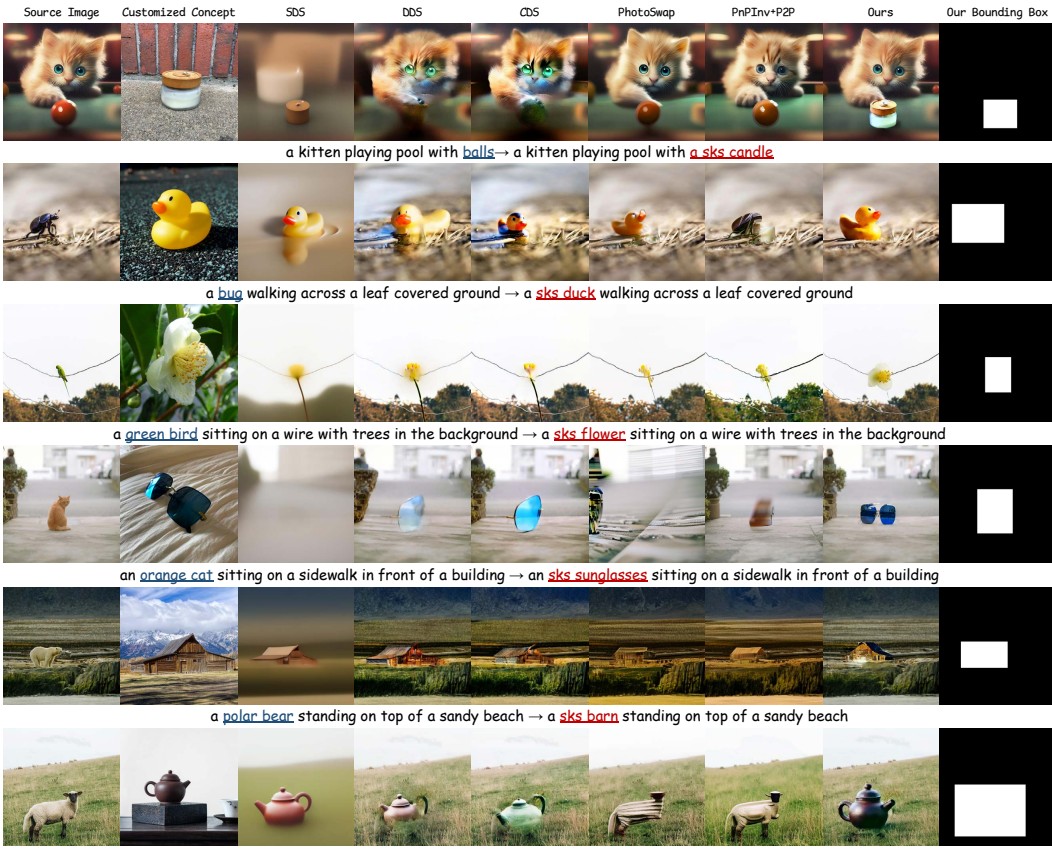

Figure 26: Visualization of the bounding boxes used in Fig. 6.

## R    LIMITATION AND FUTURE WORKS

Although the features of the entire image have been fully interacted within the self-attention layers, the base diffusion model (e.g., Stable Diffusion (Rombach et al., 2022)) is unable to accurately perceive the relative sizes of specific concepts. We visualize this phenomenon in Fig. 27: the first row shows an excessively large cup, the second row depicts a toy duck as large as the little girl, and the third row displays a pair of sunglasses nearly as large as the table. Similar issues occur in all compared methods. Nevertheless, our method still achieves the best concept swapping results.

We hope our INSTANTSWAP can inspire future research, particularly in efficiently managing concept swapping with obvious shape variance. Future work could focus on (1) extending image-based customized concept swapping to the video domain; (2) developing metrics that more accurately and comprehensively reflect the characteristics of the customized concept swapping task; and (3) achieving more lightweight and precise concept swapping.

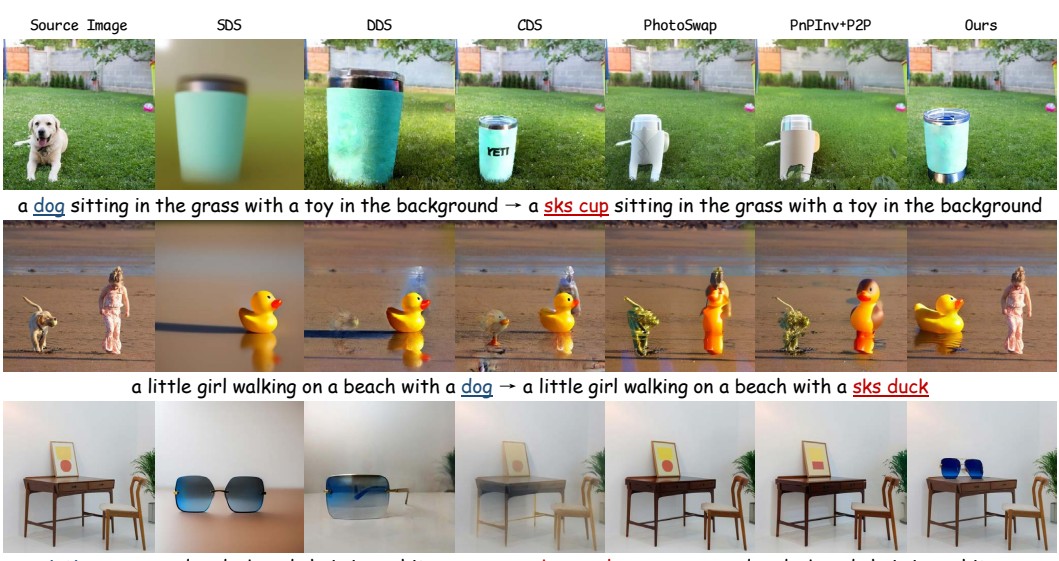

Figure 27: Failure cases.

## S    POTENTIAL SOCIAL IMPACT

The purpose of our approach is to provide users with a customized swapping tool that can perform efficient and precise customized concept swapping. However, the outstanding performance of our method also presents certain risks, including the potential for malicious third parties to create deceptive fake images. These risks are not unique to our approach; all image editing methods face similar challenges. Thanks to (Sun et al., 2023), more and more studies are focusing on detecting images that have been altered by generative models, which can prevent the misuse of related methods.

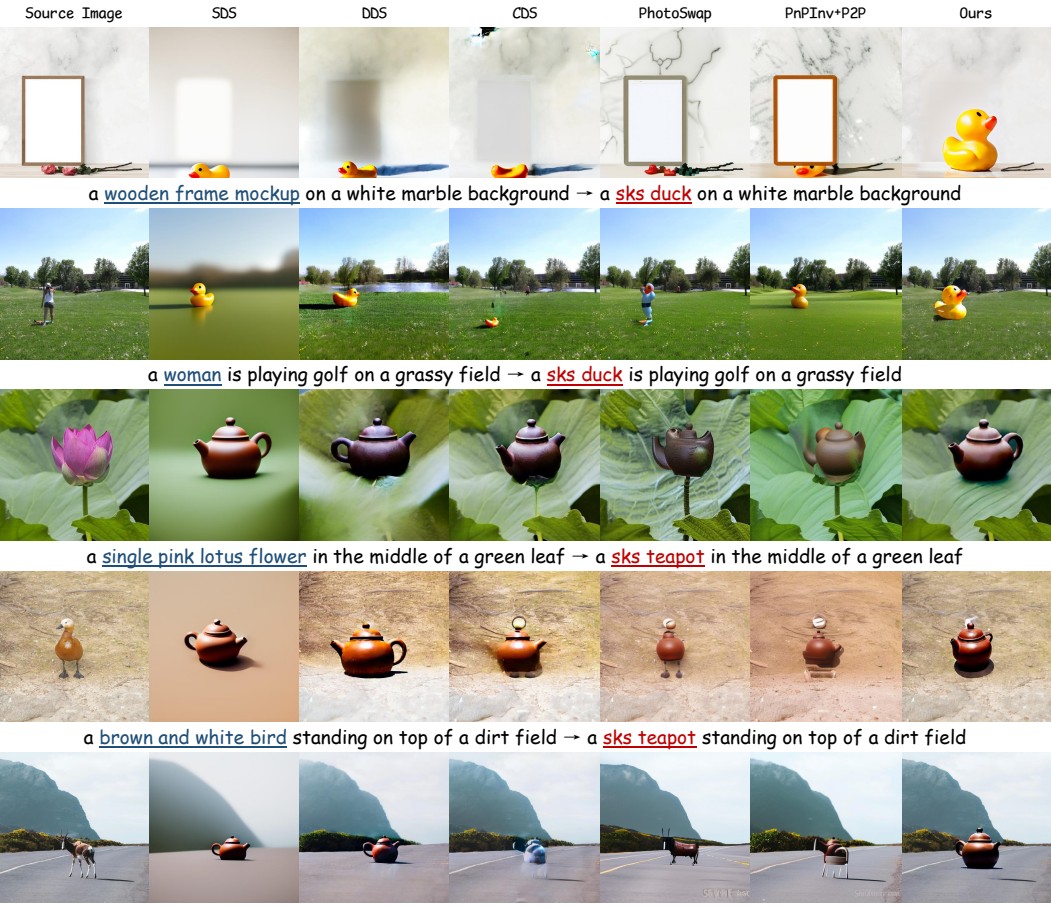

Figure 28: More qualitative comparisons.

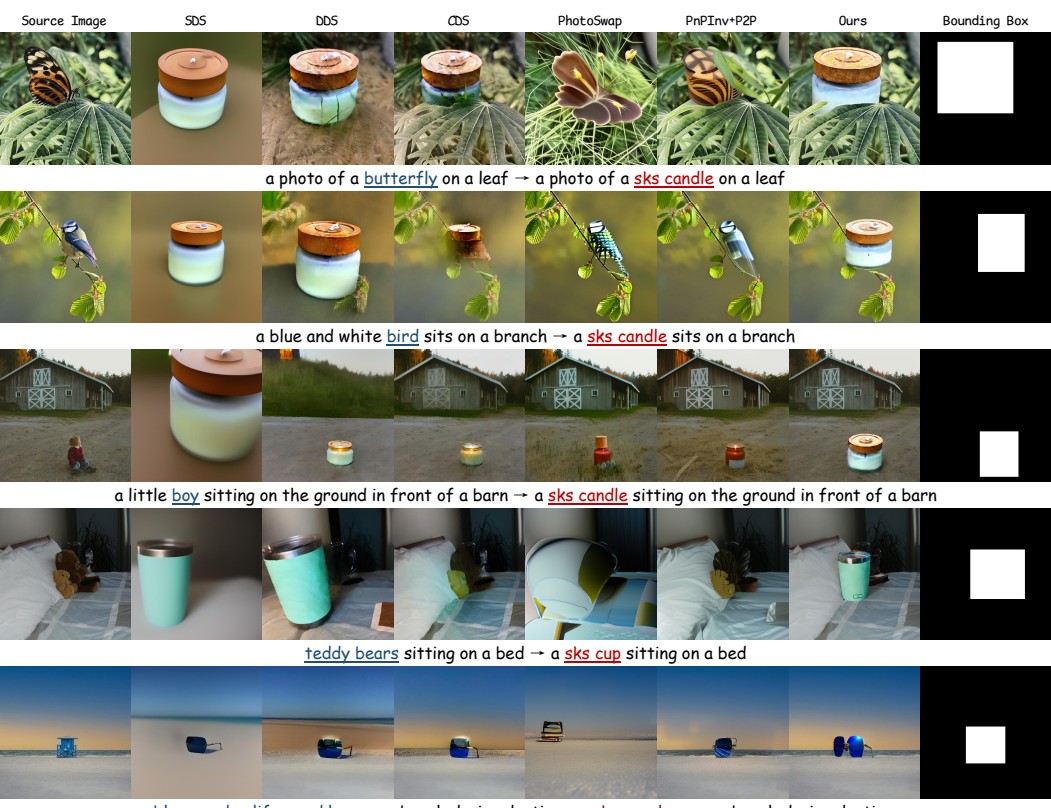

Figure 29: More qualitative comparisons.

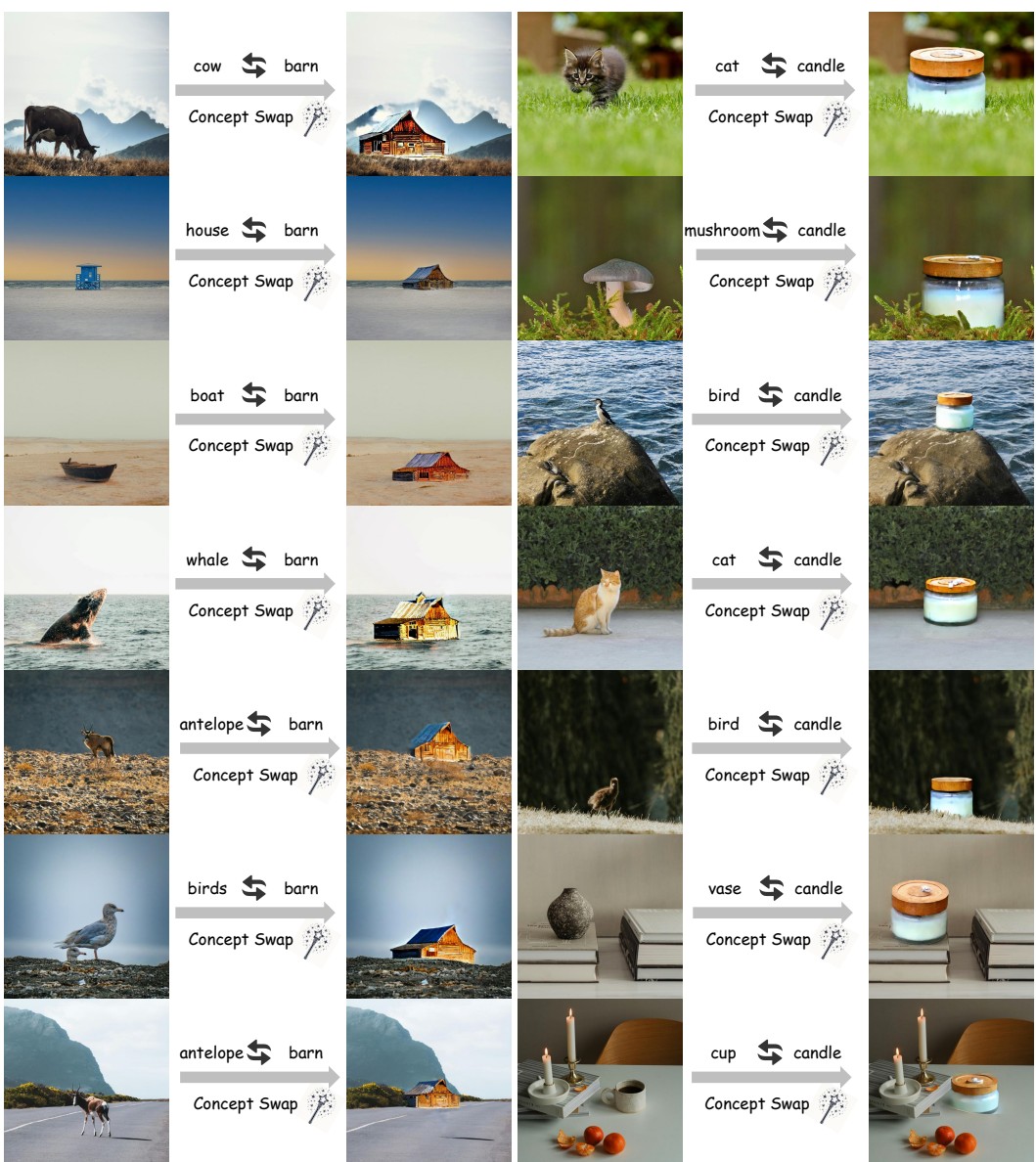

Figure 30: More qualitative results of the barn and candle.

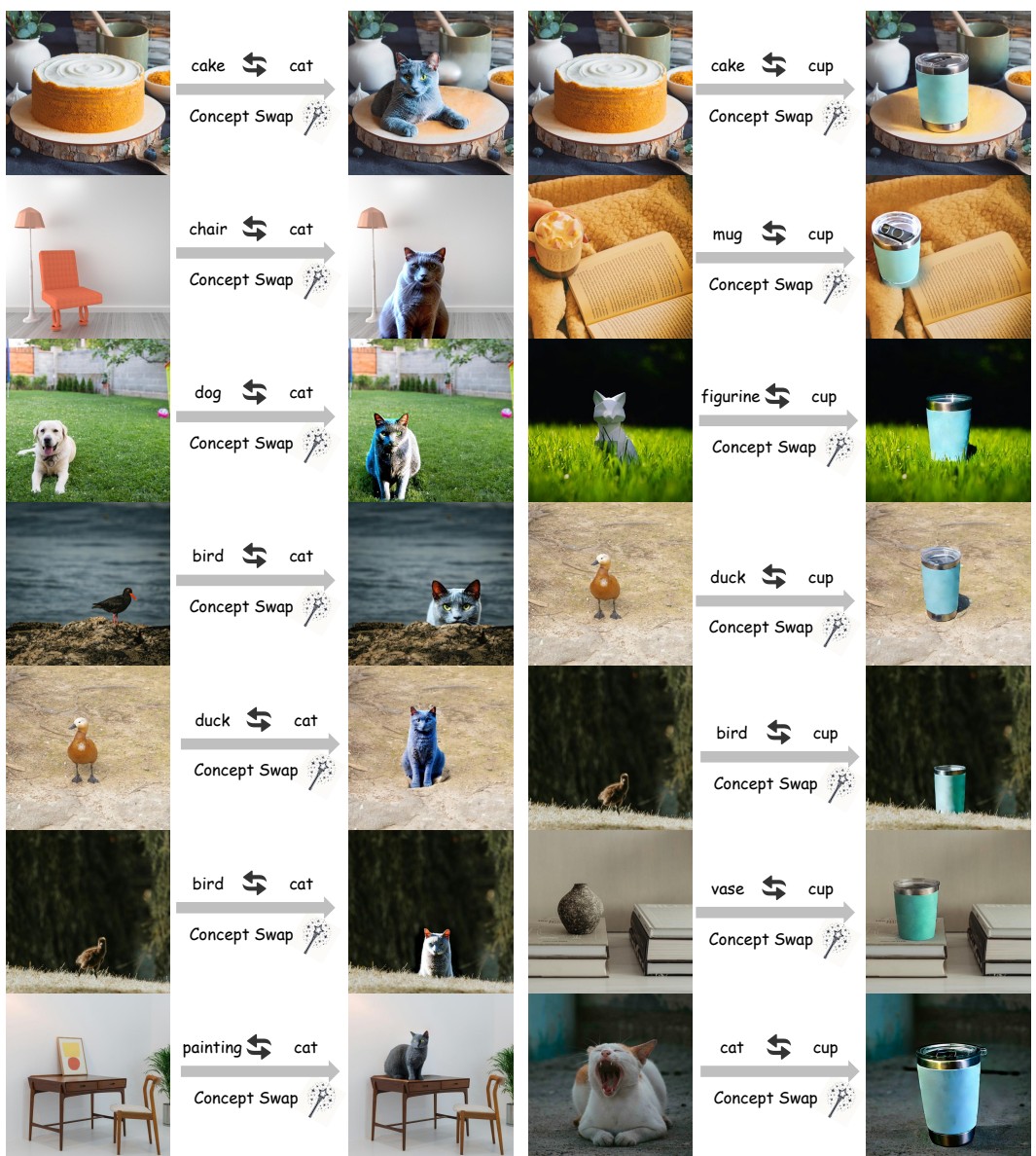

Figure 31: More qualitative results of the cat and cup.

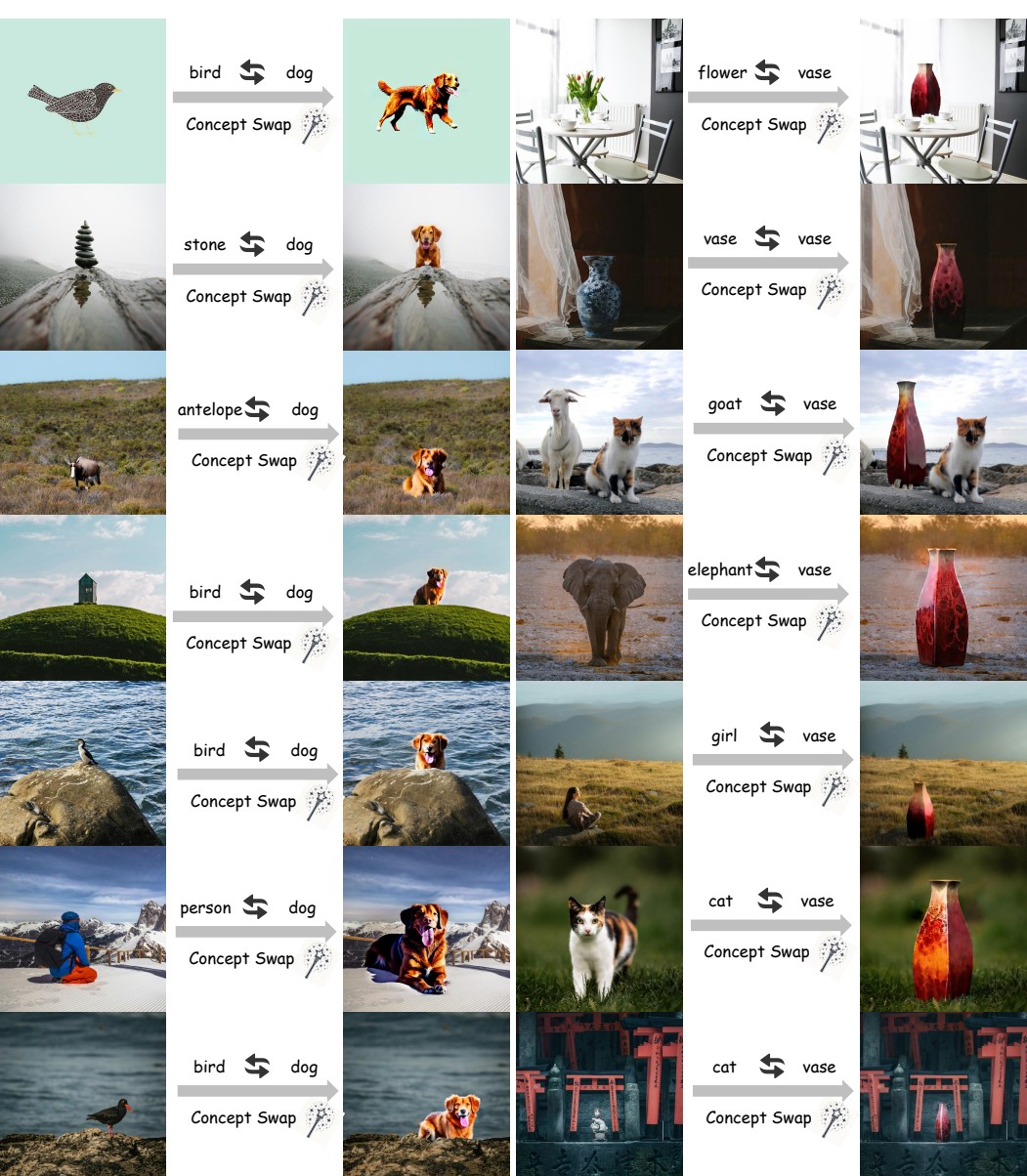

Figure 32: More qualitative results of the dog and vase.

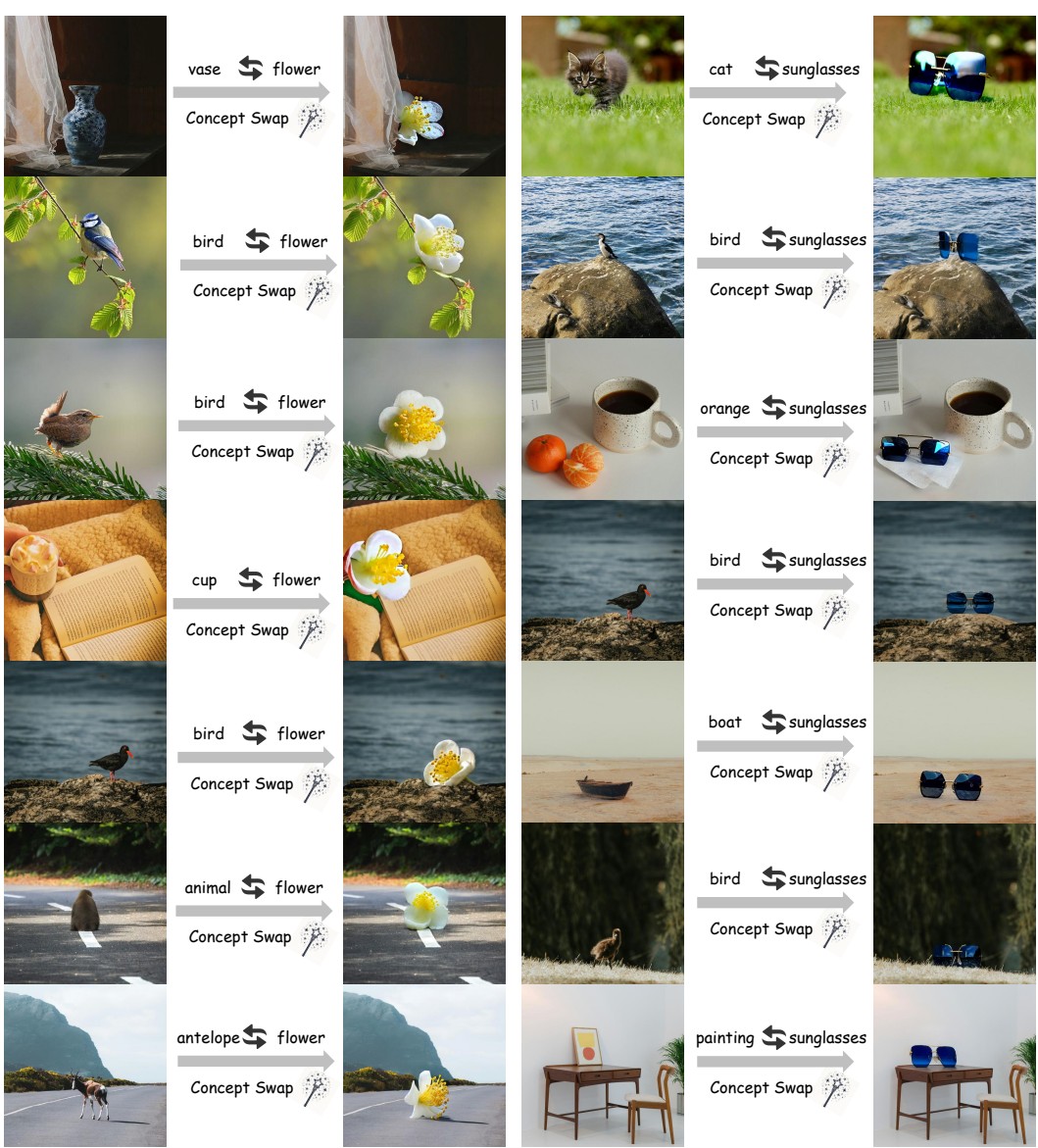

Figure 33: More qualitative results of the flower and sunglasses.

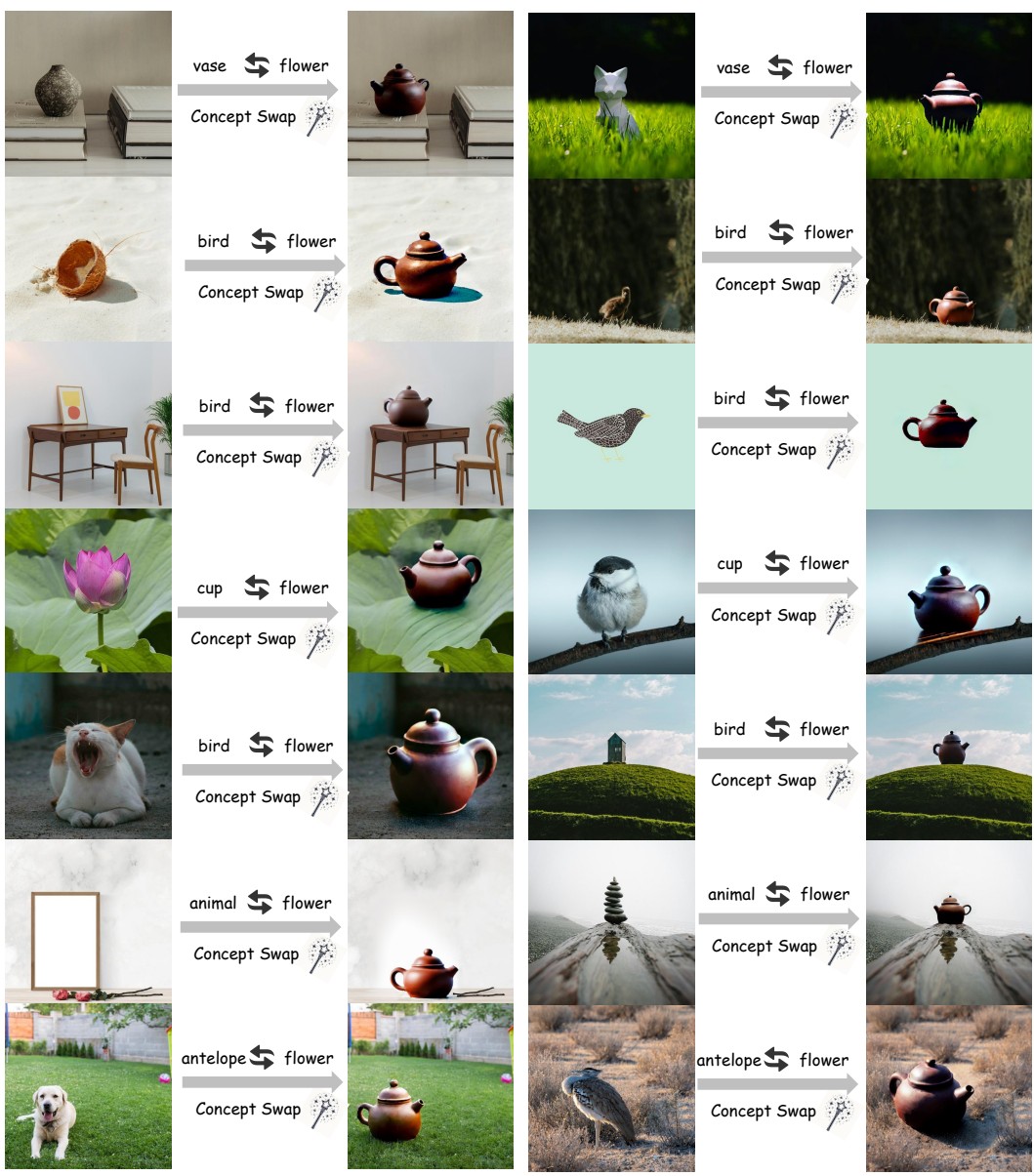

Figure 34: More qualitative results of the teapot.

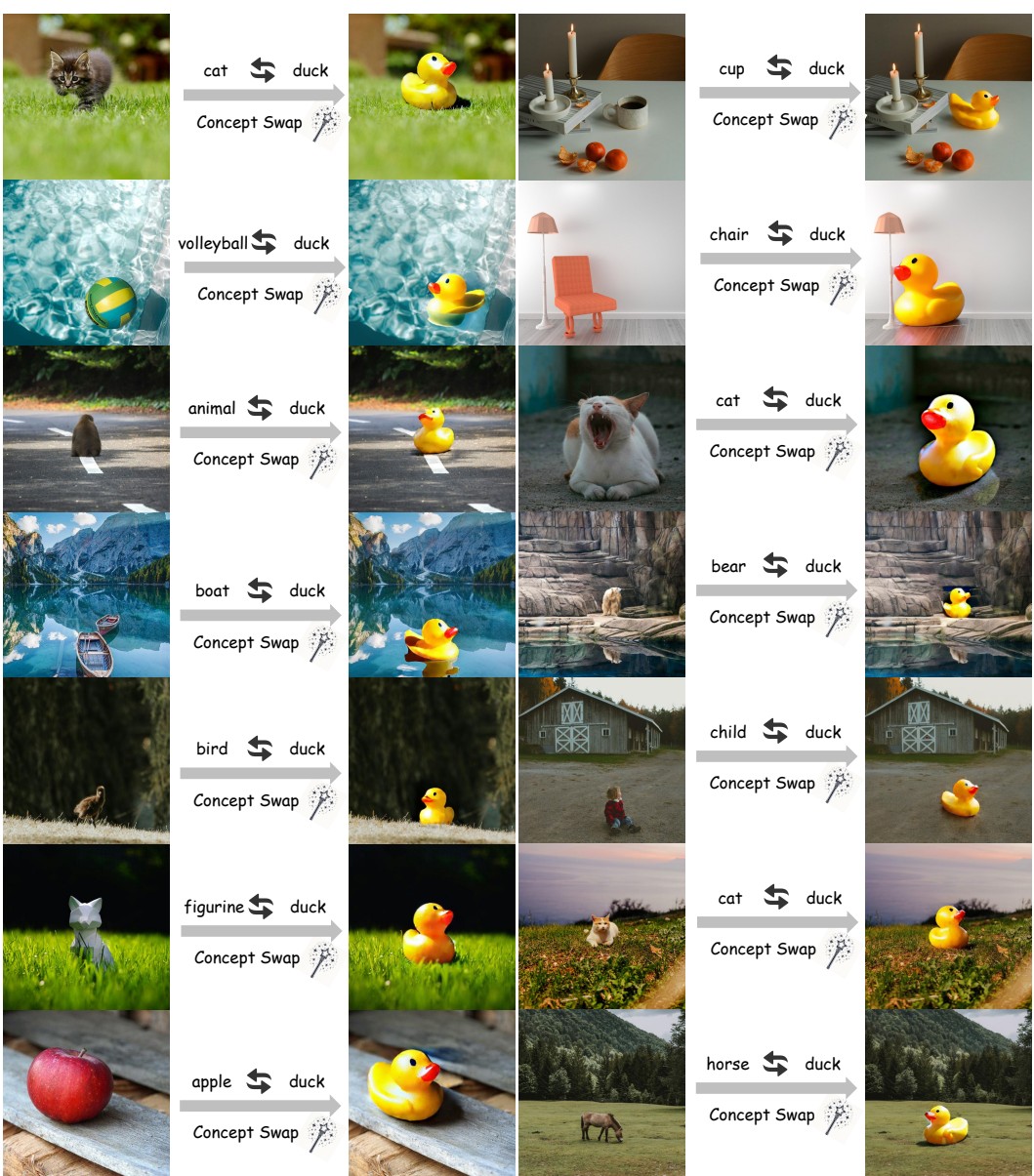

Figure 35: More qualitative results of the duck.

