# OpenReview forum: "InstantSwap: Fast Customized Concept Swapping across Sharp Shape Differences"
_ICLR.cc/2025/Conference — ICLR 2025 Poster_

### Official Review · Reviewer_VFJh · 2024-11-03

**Soundness:** 2
**Presentation:** 2
**Contribution:** 2
**Rating:** 6
**Confidence:** 4

**Summary:**

This paper proposes a training-free customized concept swapping framework. It derived bounding boxes from the Attention to map to help preserve the background information during optimizing the latents. The gradient is updated periodically for a better tradeoff between quality and inference time. A semantic enhanced module is further proposed to improve foreground consistency. Both quantitative and qualitative experiments are conducted to validate the effectiveness of this approach.

**Strengths:**

- The experimental results are comprehensive and promising.

- A benchmark dataset designed for CCS task is proposed.

- The writing is fluent and easy to understand.

**Weaknesses:**

- It would be better to give an introduction on the customization methods. This introduction could also help readers to understand the difference brought by integrating customization method into image editing.

- Since this approach needs to modify the cross attention, I wonder if it could be applied to the DiT-based architectures, like SD3.

- It seems that the semantic-enhanced operation enhances the semantic of the source/target object while mitigates the object’s interaction with the surrounding objects and background. Will it make the image unnatural?

- I am curious why P2P fails completely at this task. From the results displayed in the paper of P2P, I would expect P2P to be able to fulfill the task of changing objects. Is it because the customization method does not fit well with P2P?

**Questions:**

Please see the questions in weakness section.

---

> ### Author Response · Authors · 2024-11-22
> **Response to Reviewer VFJh (1/2)**
>
> We sincerely thank the reviewer for their insightful comments and recognition of our work. We greatly appreciate the acknowledgment of *our comprehensive and promising experimental results, fluent writing, and the benchmark we constructed*. We have refined the paper, incorporated additional experiments, and clarified the following points in the revised version. The changes relevant to your concerns are marked in **brown** and we will remove these colors in the final version.
>
> ## **W1: Detailed introduction on customization methods**
>
> Thanks for pointing this out. We have added a detailed introduction of the customization methods we used in **Section 3.2.3 and Appendix L**.
>
> * Specifically, in the target branch of our method, we first convert the target concept into semantic space with `DreamBooth`. We use a specific rare token (e.g., `sks`) to represent the concept.
>
> * During concept swapping, we can use this rare token in the target prompt to semantically represent our target customized concept and perform the customized concept swapping.
>
>
> ## **W2: Apply our method to DiT-based architectures**
>
> * Thank you for your illuminating suggestions. Our SECR enhances cross-attention maps of concepts using semantic information. This technique can be applied to any generative model with a cross-attention layer, regardless of architecture, to enrich semantic information in regions of interest.
>
> * Since there is *no explicit cross-attention layer* in SD3 [4] and it only contains a *self-attention layer*, we have included a detailed analysis in **Appendix M** to explore the architecture of SD3 and investigate how our SECR could be applied to it.
>
> * We focus on the interaction between images and text within SD3. The results indicate that the attention layer in SD3 can be decomposed into a *cross-attention operation* between image and text and a *self-attention operation* of the image or text. Our SECR can be applied to the cross-attention part of the attention layer in SD3 to enhance semantic information in regions of interest.
>
> * However, since SD3 is trained based on *Rectified Flow*, existing score distillation methods (e.g., SDS [5], DDS [6]) cannot be directly integrated with it. In future work, we will continue to explore this potential direction.
>
> [4] Scaling Rectified Flow Transformers for High-Resolution Image Synthesis. In ICML, 2024.
>
> [5] Dreamfusion: Text-to-3d using 2d diffusion. In ICLR, 2023.
>
> [6] Delta denoising score. In ICCV, 2023.
>
>
> ## **W3: Impact of semantic-enhanced operation on the naturalness of images**
> We employ the semantic-enhanced operation on foreground concepts within the cross-attention layers to derive an enhanced representation of the source and target objects. This enhanced representation effectively interacts with the features of surrounding objects and the background in the self-attention layers, mitigating inconsistencies and unnaturalness in the target image.
>
> We have provided detailed results in  **Appendix N**. The qualitative results in Fig. 23 show that:
>
> * Our method *seamlessly inpaints* objects in the background that the foreground concept occludes (upper left examples).
> * It generates *reflections* of the target concept in water (upper right examples).
> * It accurately places the target concept on *background objects* (lower left examples).
> * It also generates *natural shadows* of the target concept that are consistent with the *environmental lighting angles* (upper left and lower right examples).
>
> Moreover, considering that image naturalness is a *subjective human assessment*, we use the human preference metric [7] to quantify the naturalness of images. A higher score indicates a better alignment with human preferences. The results show that our method is the only one that receives a positive human evaluation score, demonstrating a significant advantage over other methods.
>
> | Method           | Human Preference Score ↑ |
> |------------------|:---------------------------:|
> | SDS              | -1.41                     |
> | DDS              | -0.21                     |
> | CDS              | -0.01                     |
> | PhotoSwap        | -0.86                     |
> | PnPInv+P2P       | -0.14                     |
> | Ours             | **0.51**                      |
>
> [7] Imagereward: Learning and evaluating human preferences for text-to-image generation. In NIPS, 2023.

---

> > ### Author Response · Authors · 2024-11-22
> > **Response to Reviewer VFJh (2/2)**
> >
> > ## **W4: P2P in customized concept swapping**
> >
> > **P2P seems to be able to fulfill the task of changing objects?**
> >
> > * P2P mainly focuses on *synthesized image editing* and struggles with *real image editing*. This is because P2P relies on inverting the source image into the noise space. This inversion results in *significant distortion* for real images as stated in the second paragraph of the Conclusion in the original P2P paper [8]:
> > > "The current inversion process results in a visible distortion over some of the test images."
> >
> > * In Fig. 11 of the P2P paper, P2P struggles to accurately *reconstruct* real images even with DDIM inversion, making effective editing even more difficult.
> >
> > **Does P2P fit well with customization methods?**
> >
> > * P2P can fit well with customization methods. One of our baselines, `PhotoSwap` [9], combines P2P and `DreamBooth`.
> >
> > * `PhotoSwap` adopts `null-text inversion` [10] to mitigate inversion distortion in real-image scenarios. It can effectively handle customized concept swapping without obvious shape differences.
> >
> > **Why P2P fails completely at this task?**
> >
> > * P2P itself and P2P-based methods (e.g., `PhotoSwap` and `PnPInv` [11] + P2P in our paper) heavily rely on self and cross attention maps to maintain background consistency. Although this improves their background consistency, it limits their ability to change objects with *shape differences*.
> >
> > * However, in the task of customized concept swapping, the target concept is highly customized and naturally has a *sharp shape difference* from the source concept. This presents a significant challenge for P2P-based methods.
> >
> > * As shown in Fig.6 in our paper, P2P-based methods can only edit within the existing shape and cannot change the shape of the source concept, leading to their failure in this task.
> >
> > [8] Prompt-to-prompt image editing with cross attention control. In ICLR, 2023.
> >
> > [9] Photoswap: Personalized subject swapping in images. In NIPS, 2023.
> >
> > [10] Null-text inversion for editing real images using guided diffusion models. In CVPR, 2023.
> >
> > [11] Pnp inversion: Boosting diffusion-based editing with 3 lines of code. In ICLR, 2024.

---

> ### Author Response · Authors · 2024-11-27
> **A Gentle Reminder of the Final Feedback**
>
> **Dear Reviewer VFJh:**
>
> Thank you once again for dedicating your valuable time to reviewing our paper and providing constructive comments! As the end of the discussion period approaches, we kindly ask if our responses have satisfactorily addressed your concerns. If you have any further inquiries about our method, or if you still have any concerns about our paper, please do not hesitate to inform us. We are more than willing to engage in timely discussions with you.
>
> Sincerely,
>
> The Authors

---

> ### Author Response · Authors · 2024-11-29
> **A Second Reminder of the Post-rebuttal Feedback**
>
> **Dear Reviewer VFJh:**
>
> We would like to thank you again for your valuable feedback on our paper.
>
> As the period for the Author-Reviewer discussion is closing very soon, we would like to use this opportunity to kindly ask if our responses sufficiently clarify your concerns. We sincerely appreciate your time and consideration.
>
> Best Regards,
>
> The Authors

---

> ### Author Response · Authors · 2024-12-01
> **The Third Warm Reminder of the Post-rebuttal Feedback**
>
> **Dear Reviewer VFJh:**
>
> We notice that all other reviewers have posted their post-rebuttal comments to our response but we still have not yet received any further information from you. We greatly appreciate your initial comments and fully understand that you may be extremely busy at this time.
>
> As the deadline for the Reviewer-Author discussion phase is fast approaching (**less than two days left**), we respectfully ask whether we have addressed your questions and concerns adequately. Your feedback would be greatly appreciated, and we would be delighted to engage in further discussions if needed.
>
> Best Regards,
>
> The Authors

---

> > ### Comment · Reviewer_VFJh · 2024-12-01
> > **Thanks for response**
> >
> > Most of my concerns have been addressed in author responses. Thus I will increase my rating.

---

> > > ### Author Response · Authors · 2024-12-02
> > > **Thanks to Reviewer VFJh**
> > >
> > > Thanks for your response. We appreciate your engagement in our discussion and your recognition of our efforts!

---

### Official Review · Reviewer_iqew · 2024-11-05

**Soundness:** 3
**Presentation:** 3
**Contribution:** 3
**Rating:** 6
**Confidence:** 4

**Summary:**

Recent advances in Customized Concept Swapping (CCS) enable text-to-image models to swap concepts, but existing methods struggle to maintain foreground and background consistency, particularly with large shape disparities, and often require time-intensive processes. InstantSwap addresses these challenges by using bounding box analysis and cross-attention mechanisms to enhance both foreground and background consistency while limiting modifications to the background and enhancing foreground focus. This method reduces computation time by periodically calculating gradients, enhancing efficiency with minimal performance loss, and extensive evaluations show InstantSwap's effectiveness and adaptability.

**Strengths:**

The motivation is clear.

The results appear promising and solid.

The experiments are thorough.

The writing is easy to follow.

**Weaknesses:**

For each concept replacement, the method first needs to train a DreamBooth model and then perform score distillation, which is time-intensive.

Both the source and reference branches use DreamBooth-tuned UNet. It would be beneficial to validate the method using text inversion to demonstrate its generalization capability.

What about the failure cases?

It's interesting that the method can handle concepts with significant shape changes. If the original image's concept is very small, resulting in a small bounding box, how does the target image’s foreground region expand without additional processing?

**Questions:**

Please see the weakness

---

> ### Author Response · Authors · 2024-11-22
> **Response to Reviewer iqew**
>
> We sincerely thank the reviewer for their insightful comments and recognition of our work, especially for acknowledging our *comprehensive experiments, solid results, clear motivation, and smooth writing*. We have refined the paper, incorporated additional experiments, and clarified the following points in the revised version. The changes relevant to your concerns are marked in **blue** and we will remove these colors in the final version.
>
> ## **W1: Customization is time-intensive**
>
> * The overall time added is very mild. As shown in the table below, `DreamBooth` requires about *8.5 minutes* to learn a single concept. Meanwhile, our method can be combined with other more efficient customization methods such as `Custom Diffusion` [2] which only requires about *3.5 minutes* for each concept.
>
> |                | DreamBooth         | Custom Diffusion    |
> |:--------------:|:------------------:|:-------------------:|
> | Training Time  |      ~8.5min       |       ~3.5min       |
>
> We have added the above discussion in **Appendix J**. The visual results of our method combined with `Custom Diffusion` can be found in Fig.22.
>
> [2] Multi-concept customization of text-to-image diffusion. In CVPR, 2023.
>
> ## **W2: Combine our method with Textual Inversion**
>
> Thanks for your suggestion. Our method can indeed be combined with other customization methods such as `Textual Inversion` (TI) [3]. We have included results in **Appendix K** and present our quantitative results here. Please refer to our modified manuscript for more qualitative results and analysis.
>
> |Method| CLIP-I ↑ | PSNR ↑ | LPIPS ↓ | MSE ↓ | SSIM ↑ | CLIP-T ↑ | Time ↓ |
> |---------------|----------|--------|---------|-------|--------|----------|--------|
> | Ours          | 75.00    | 27.39  | 47.68   | 27.87 | 86.58  | 25.74    | 19.83  |
> | Ours with TI  | 72.20    | 28.16  | 45.91   | 24.20 | 86.76  | 24.57    | 19.92  |
>
> * The results show that our method integrates well with `Textual Inversion`, achieving similar performance in background consistency and inference time.
> * Due to the limited customization capabilities of `Textual Inversion`, its foreground consistency is not as high as with our original method.
>
> [3] An image is worth one word: Personalizing text-to-image generation using textual inversion. In ICLR, 2023.
>
> ## **W3: More failure cases**
>
> We have presented failure cases in the **limitations part of Appendix P**. We have added more failure cases to the modified manuscript and conducted a detailed analysis. Specifically, due to the limited capability of the base diffusion model (e.g., `Stable Diffusion`), it is unable to perceive the relative sizes between specific concepts. We visualize this phenomenon in Fig. 25:
>
> * First row: an extremely large cup.
>
> * Second row: a toy duck the size of a little girl.
>
> * Third row: a pair of sunglasses nearly as large as the table.
>
> Similar issues occur in all compared methods. Nevertheless, our method still achieves the best concept swapping results.
>
>
> ## **W4: How to expand the foreground region**
>
> * The goal of Customized Concept Swapping is to smoothly replace the foreground object with the customized object. We care about (1) whether the foreground region is re-occupied with the customized object and (2) whether the transition between the foreground and background remains natural and smooth. As a 2D image editing task, whether the physical size of an object is consistent with the surrounding environment (highly dependent on the perspective angle) is not a focus in this field.
>
> * As we mentioned in the **limitation part of Appendix P**, perceiving the relative size of specific concepts is very challenging for the current base diffusion model. Previous image editing methods also struggle to effectively manage this issue.
>
> * If we really want to handle this scenario, we can expand our obtained bounding by adjusting the threshold $\beta$ adaptively during the swapping process to make the bounding box larger, thereby expanding the foreground region.

---

> ### Author Response · Authors · 2024-11-27
> **A Gentle Reminder of the Final Feedback**
>
> **Dear Reviewer iqew:**
>
> Thank you once again for dedicating your valuable time to reviewing our paper and providing constructive comments! As the end of the discussion period approaches, we kindly ask if our responses have satisfactorily addressed your concerns. If you have any further inquiries about our method, or if you still have any concerns about our paper, please do not hesitate to inform us. We are more than willing to engage in timely discussions with you.
>
> Sincerely,
>
> The Authors

---

> ### Author Response · Authors · 2024-11-29
> **A Second Reminder of the Post-rebuttal Feedback**
>
> **Dear Reviewer iqew:**
>
> We would like to thank you again for your valuable feedback on our paper.
>
> As the period for the Author-Reviewer discussion is closing very soon, we would like to use this opportunity to kindly ask if our responses sufficiently clarify your concerns. We sincerely appreciate your time and consideration.
>
> Best Regards,
>
> The Authors

---

> > ### Comment · Reviewer_iqew · 2024-11-30
> >
> > Thanks authors for responses. My concerns have been addressed so I kept my acceptance score.

---

> > > ### Author Response · Authors · 2024-12-01
> > > **Thanks to Reviewer iqew**
> > >
> > > Thanks for your response. We appreciate your engagement in our discussion and your recognition of our efforts!

---

### Official Review · Reviewer_nPYy · 2024-11-06

**Soundness:** 3
**Presentation:** 2
**Contribution:** 3
**Rating:** 6
**Confidence:** 3

**Summary:**

This paper proposes InstantSwap, a training-free framework for Customized Concept Swapping(CCS). CCS works on transfers the target concept described by target images and target prompt to the location of source concept in the source image. This paper utilizes the cross attention map and the self-attention map of U-Net in diffusion model for source image and source prompt to extract bounding box of the source concept automatically. Then they apply the bounding box to filter the gradients in background from a refined SDS loss. In this way, they can achieve an optimization preserving the background information. To emphasize the concepts in the images, they also use the semantic information of corresponding prompts and the estimated bbox to augment representation of concepts. Additionally, this paper also presents a step-skipping gradient update strategy which reuse previous gradients for current iteration to increase the inference speed. Experiments present the advantages of the proposed method over previous works.

**Strengths:**

1. The proposed method presents a complete pipeline for improve effectiveness and efficiency in Customized Concept Swapping(CCS) task.
2. The proposed method obtains state-of-the-art performance compared to previous works.
3. This paper also contributes benchmark for Customized Concept Swapping(CCS) task.

**Weaknesses:**

1. Theoretical analysis about why we can directly apply mask on gradient computing is missing. Masking will produce a distribution shift, why it can converge to a reasonable solution requires some analysis.
2. How combining self-attention and cross-attention for automatic bbox generation affects the performance seems not be discussed.
3. How the number of target images affect the performance is not mentioned.

**Questions:**

1. How does the proposed method work for multi-object scenario?

---

> ### Author Response · Authors · 2024-11-22
> **Response to Reviewer nPYy (1/2)**
>
> We sincerely thank the reviewer for their insightful comments and recognition of our work, particularly for acknowledging the *integrity*, *effectiveness*, and *efficiency* of our method. We have refined the paper, incorporated additional experiments, and clarified the following points in the revised version. The changes relevant to your concerns are marked in **red** and we will remove these colors in the final version.
>
> ## **W1: Theoretical analysis of background gradient masking (BGM)**
>
> Thanks for your suggestion.
> * As shown in Fig. 3 of our paper, a complete optimization step of our method consists of two stages: *forward pass and backward propagation*. During the forward pass, we input the source and target images and compute gradient $\nabla_{z} \mathcal{L}$ as $\nabla_z \mathcal{L}=w(t)\left(\epsilon_\phi\left(z_t, t, \tau\left(P_t\right)\right)-\hat{\epsilon}_\phi\left(\hat{z_t}, t, \tau\left(P_s\right)\right)\right.$  (Eq. 10 of the main paper). *No mask* is applied during gradient computing, allowing sufficient interaction between the foreground and background within the **self-attention layer**. This ensures that *no distribution shift* occurs between the foreground and background in the gradient $\nabla _{z} \mathcal{L}$.
>
> * Furthermore, the gradient $\nabla_{z} \mathcal{L}$ shares the same dimensions as the image latent $z$ (see more gradient visual results in Fig. 18 of the Appendix). The backward propagation of the gradient $\nabla_{z} \mathcal{L}$ is a *pixel-wise* update on $z$. Based on this, we directly apply the mask on the gradient of the background pixels in $\nabla_{z} \mathcal{L}$ before backward propagation, resulting in our $\nabla_{z} \mathcal{L}_{BGM}$.
>
> * The update of our gradient on the foreground is the *same* as that of $\nabla_{z} \mathcal{L}$ on the foreground. Since there is no distribution shift in $\nabla_{z} \mathcal{L}$, our $\nabla_{z} \mathcal{L}_{BGM}$ similarly avoids introducing distribution shifts in the foreground update.
>
> * Besides, as shown in Fig. 23 in the Appendix, our method consistently generates results that are natural and coherent in both the foreground and background, further proving that there is no distribution shift between the foreground and background in our $\nabla_{z} \mathcal{L}_{BGM}$.
>
> * If any explanation remains unclear, please do not hesitate to reach out. We are very happy to clarify and address any misunderstandings.

---

> > ### Author Response · Authors · 2024-11-22
> > **Response to Reviewer nPYy (2/2)**
> >
> > ## **W2: Ablation study on the automatic bbox generation**
> >
> > We set the element-wise exponentiation of self-attention map $A^s$ and cross-attention map $A^c$ in Eq.8 to $\alpha_s$ and $\alpha_c$ respectively and conduct an ablation study on the combination of self-attention and cross-attention. We have added the results in **Appendix H** and present our quantitative results here. For more qualitative results and analysis, please refer to our modified manuscript. Specifically, we divide their combination into the following scenarios:
> >
> > * $\alpha_s=0$, $\alpha_c=1, 2$: In this case, $A^s$ is an *all-ones* matrix, so each element in $\hat{A}^c$ has the same value, which is equal to the sum of all elements in $A^c$. After normalization and applying a threshold, the entire bounding box is activated.
> >
> > * $\alpha_s=1$, $\alpha_c=0$: In this case, $A^c$ is a vector of all ones, so each item in $\hat{A^c}$ is the sum of the elements in the corresponding row of $A^s$. Since $A^s$ is the *output of softmax*, the sum of each row's elements in $A^s$ is 1. Therefore, $\hat{A^c}$ is also a matrix of all ones, which ultimately results in the entire bounding box being activated.
> >
> > * $\alpha_s=1$, $\alpha_c=1$: In this case, the foreground region (FR) in the image cannot be highlighted, resulting in a larger bbox, which causes the background of the source image to be unnecessarily altered.
> >
> > * $\alpha_s=2$, $\alpha_c=0$: In this situation, using the self-attention map alone cannot effectively highlight the FR, resulting in a very imprecise bounding box.
> >
> > * $\alpha_s=2$, $\alpha_c=1, 2$: In this case, the self-attention map takes a leading role in the bbox generation process, producing a smaller bbox, thereby reducing foreground consistency.
> >
> > * $\alpha_s=1$, $\alpha_c=2$ (Ours): Our setting achieves a proper balance between foreground and background consistency, fully covering the source concept while minimizing background modification.
> >
> > | $\alpha_s$, $\alpha_c$ | Bbox | CLIP-I ↑ | PSNR ↑ | LPIPS ↓ | MSE ↓  | SSIM ↑ | CLIP-T ↑ |
> > |------------------------|------------------------|----------|--------|---------|-------|--------|----------|
> > | 0, 1                   | full image             | 72.70    | 15.25  | 327.23  | 350.01| 64.80  | 21.23    |
> > | 0, 2                   | full image             | 72.70    | 15.25  | 327.23  | 350.01| 64.80  | 21.23    |
> > | 1, 0                   | full image             | 72.70    | 15.25  | 327.23  | 350.01| 64.80 | 21.23    |
> > | 1, 1                   | larger than FR         | 74.18    | 23.83  | 93.12   | 82.44 | 82.95  | 25.19    |
> > | 1, 2 (Ours)  | **Properly cover FR**      | **75.00**    | 27.37  | 48.40   | 28.03 | 86.58  | **25.74**    |
> > | 2, 0                   | larger than FR         | 73.66    | 26.49  | 65.56   | 56.81 | 84.81  | 24.97    |
> > | 2, 1                   | smaller than FR        | 73.67    | 30.08  | 37.34   | 19.15 | 87.30  | 25.12    |
> > | 2, 2                   | smaller than FR        | 72.66    | **30.86**  | **34.38**   | **15.38** | **87.54**  | 24.78    |
> >
> >
> >
> > ## **W3: Analysis of the number of target images**
> >
> > Before concept swapping, our method utilizes `DreamBooth` and a set of images (typically fewer than 5) $\mathcal{X} _t=\\{ x _i \\} _{i=1}^M$ for customizing certain concepts, where $M$ is the number of target images used in the customization process.
> >
> > We have added a detailed analysis of the number of target images $M$ in **Appendix I** and present our quantitative results here. Please refer to our modified manuscript for more detailed qualitative results and analysis.
> >
> > |        | $M=1$  | $M=2$  | $M=3$  | $M=4$  |
> > |--------|----------|----------|----------|----------|
> > | CLIP-I ↑ | 74.78  | 75.40  | 75.57    | **75.65**   |
> > | CLIP-T ↑ | 28.94   | 28.93    | 29.13    | **29.44**   |
> >
> > ## **Q1: How can InstantSwap handle multi-object scenario**
> >
> > * As shown in **Sec 4.6 Multi-concept swapping** of the main paper, our method can perform multi-object swapping by *sequentially performing multiple single-concept swaps*. We also provide a visual example in Fig. 10.
> >
> > * Besides this simple strategy, we can employ multiple bounding boxes and implement our SECR strategy simultaneously on multiple objects within an image. Furthermore, combining our method with `Gligen` [1] can facilitate handling multi-object scenarios with implicit grounding conditions. We will explore this as a future work.
> >
> > [1] Gligen: Open-set grounded text-to-image generation. In CVPR, 2023.

---

> ### Author Response · Authors · 2024-11-27
> **A Gentle Reminder of the Final Feedback**
>
> **Dear Reviewer nPYy:**
>
> Thank you once again for dedicating your valuable time to reviewing our paper and providing constructive comments! As the end of the discussion period approaches, we kindly ask if our responses have satisfactorily addressed your concerns. If you have any further inquiries about our method, or if you still have any concerns about our paper, please do not hesitate to inform us. We are more than willing to engage in timely discussions with you.
>
> Sincerely,
>
> The Authors

---

> > ### Comment · Reviewer_nPYy · 2024-11-28
> >
> > I acknowledge I have read authors' responses and the reviews from other reviewers. Most of my concerns are addressed. The theoretical analysis doesn't look convincing but I understand it is not the main focus of this paper. After reading other reviews, I have some new comments:
> > 1. For a training-free method, it is important to prove its generalizability on different architecture. Although the proposed method doesn't work for Stable Diffusion 3, it would also be better to present some results for other stable diffusion models like stable diffusion 1.5 and stable diffusion XL.
> > 2. I don't think the failure case and the limitation is fully discussed. According to Figure 28 and Figure 29, InstantSwap seems to be limited to handle concepts with complicated texture. For example, the details of barn look different for different images and the identity of cat is not preserved.
> > 3. The proposed benchmark ConceptBench only provides 10 concepts and most of concepts are covered by DreamBooth, which seems to be limited. Considering the proposed method builds upon pre-trained DreamBooth, more challenging unseen concepts might be necessary.

---

> ### Author Response · Authors · 2024-11-29
> **Thank you for the response**
>
> We appreciate your invaluable feedback! We shall exert our utmost efforts to address your inquiries:
>
> ***
>
> ## **Q1 Adapt InstantSwap to other Stable Diffusion models**
>
> * We combine our method with `Stable Diffusion 1.5` and conduct comprehensive experiments. We present the quantitative results in the table below. We also provide qualitative results in:
> https://anonymous.4open.science/r/ICLR_InstantSwap-3030/SD1.5.pdf
>
> * The results show that our method can integrate well with `Stable Diffusion 1.5`.
>
>
> | Method | CLIP-I ↑ | PSNR ↑ | LPIPS ↓ | MSE ↓ | SSIM ↑ | CLIP-T ↑ | Inference Time (on a single A100) ↓ |
> |------------|----------|--------|---------|-------|--------|----------|------------------------------------|
> | Ours w/ SD1.5      | 74.27    | 27.98  | 47.04   | 25.24 | 86.01  | 26.10    | 15.09s |
> | Ours       | 75.00    | 27.37  | 48.40   | 28.03 | 86.58  | 25.74    | 13.38s|
>
> ***
>
> ## **Q2 Handle concepts with complicated textures**
>
> * Generating high-fidelity concepts with complicated textures is indeed a challenging problem. We analyze that it is primarily caused by customization methods like DreamBooth, which *struggles* to consistently generate complex concepts in different images.
>
> * We provide an example in the anonymous link below, where we directly use DreamBooth with the target prompt to generate images. It is evident that concepts generated by DreamBooth are different for different images.
> https://anonymous.4open.science/r/ICLR_InstantSwap-3030/more_failure_case.pdf
>
> * However, even if DreamBooth cannot effectively customize these complex concepts, our method still completes the concept swapping precisely (see qualitative results in Q3 below).
> Besides, our method can be integrated with other customization methods despite DreamBooth, which is discussed in W1 and W2 of *Reviewer iqew*.
>
> ***
>
> ## **Q3 More challenging concepts**
>
> * During the experiment, we pair each concept from ConceptBench with each source image from SwapBench, resulting in a total of 1,600 images for evaluation. This is larger than the 700 images in `PnPInv` [11] and the 55 images in `PnP` [12].
>
> * Additionally, we include more uncommon challenging concepts from `DreamBench++` [13]. Due to time constraints, we only conduct experiments on the first 50 images in SwapBench. We present the quantitative results in the table below. We also provide qualitative results in:
> https://anonymous.4open.science/r/ICLR_InstantSwap-3030/challenging_concept.pdf
>
> * The results show that these challenging concepts indeed reduce the foreground consistency of our method. Nevertheless, our approach still faithfully completes the concept swapping and surpasses all compared methods on all metrics. We will include more challenging concepts, conduct experiments on the whole SwapBench, and report the results in the final version.
>
> | Method        | CLIP-I ↑ | PSNR ↑ | LPIPS ↓ | MSE ↓  | SSIM ↑ | CLIP-T ↑ |
> |---------------|----------|--------|---------|--------|--------|----------|
> | SDS           | 67.09    | 19.47  | 307.43  | 140.39 | 73.76  | 25.16    |
> | DDS           | 64.28    | 21.73  | 130.48  | 89.82  | 81.54  | 26.28    |
> | CDS           | 65.50    | 20.45  | 139.67  | 120.78 | 80.28  | 26.56    |
> | PhotoSwap     | 59.01    | 23.54  | 139.39  | 69.09  | 79.72  | 23.66    |
> | PnPInv+P2P    | 60.70    | 24.41  | 102.76  | 51.95  | 83.59  | 25.77    |
> | Ours          | **69.13**    | **26.56**  | **54.10**   | **32.84**  | **87.53**  | **27.33**    |
>
> ***
>
> [11] Pnp inversion: Boosting diffusion-based editing with 3 lines of code. In ICLR, 2024.
>
> [12] Plug-and-Play Diffusion Features for Text-Driven Image-to-Image Translation. In CVPR, 2023.
>
> [13] Dreambench++: A human-aligned benchmark for personalized image generation. In Arxiv, 2024.
>
> ***
>
> Lastly, we thank you once again for your response. If you have any further questions, we would be delighted to continue the discussion with you.

---

> > ### Comment · Reviewer_nPYy · 2024-12-01
> >
> > Thank authors for the responses. My concerns are well addressed. I would encourage authors to consider including more challenging concepts to make the proposed benchmark compatible to more scenarios. I would like to keep my acceptance score.

---

> > > ### Author Response · Authors · 2024-12-02
> > > **Thanks to Reviewer nPYy**
> > >
> > > Thanks for your response. We appreciate your engagement in our discussion and your recognition of our efforts!

---

### Author Response · Authors · 2024-11-22
**General response**

We extend our sincere gratitude to all the reviewers (**R1-nPYy**, **R2-iqew**, and **R3-VFJh**) for their insightful and considerate reviews, which help us to emphasize the contributions of our approach.

We are very encouraged to hear that the reviewers recognized the **clear motivation** (R2) and **effectiveness** (R1, R2) of our method, our **comprehensive and solid experiments** (R2, R3), the **advantageous results** we presented (R1, R2), the **benchmark** we proposed (R1, R3), and the **fluent writing** of our paper (R2, R3).

We would also like to express our sincere gratitude to the reviewers for their insightful identification of areas where our manuscript could be strengthened.
We have taken all the suggestions carefully and updated our previous version. In the revised manuscript, we have made the following **changes** and highlighted them in different colors for different reviewers (**R1-red**, **R2-blue**, **R3-brown**):

1. Added the ablation study on automatic bounding box generation in **Appendix H**. (R1)
2. Added the analysis of the number of target images in **Appendix I**. (R1)
3. Added the discussion on the time cost of the customization process in **Appendix J**. (R2)
4. Added the results of combining our method with Textual Inversion in **Appendix K**. (R2)
5. Added more failure cases and analysis in **Appendix P**. (R2)
6. Refined **Section 3.2.3** and added a detailed introduction of the customization methods we used in **Appendix L**. (R3)
7. Added a detailed analysis of how our SECR can be applied to DiT-based architecture (e.g., SD3) in **Appendix M**. (R3)
8. Added more qualitative and quantitative results to demonstrate the naturalness of our resulting image in **Appendix N**. (R3)

We sincerely hope to **engage in further discussion with the reviewers to ensure all concerns have been fully addressed**. If any aspects of our work remain unclear, we welcome **any further feedback** to help improve our manuscript. Thank you very much again!

---

### Meta-Review · Area_Chair_xWzC · 2024-12-18

**Metareview:**

The paper proposes a training-free framework for Customized Concept Swapping(CCS). It transfers a target concept described by images and a prompt to the location of a source concept in the source image.

The paper is well written, and the method achieves state-of-the-art performance across various thorough experiments. The work additionally contributes a benchmark for Customized Concept Swapping(CCS) task.

After the post-rebuttal discussion, all reviewers unanimously agree that the paper should be accepted. Remaining minor concerns are the inclusion of more challenging concepts to the proposed benchmark. We encourage the authors to address this before submitting the camera-ready version of the paper.

Overall, this is solid work that is relevant to the community and should be presented at ICLR.

**Additional Comments On Reviewer Discussion:**

All reviewers were responsive during the rebuttal. The authors formulated an extensive rebuttal that led to a unanimous acceptance post-rebuttal, with minor concerns remaining.

---

### Decision · Program_Chairs · 2025-01-22

Accept (Poster)